# *N*-chlorination mediates protective and immunomodulatory effects of oxidized human plasma proteins

**Agnes Ulfig, Anton V Schulz, Alexandra Müller, Natalie Lupilov, Lars I Leichert***

Institute of Biochemistry and Pathobiochemistry – Microbial Biochemistry, Ruhr University Bochum, Bochum, Germany

**Abstract** Hypochlorous acid (HOCl), a powerful antimicrobial oxidant, is produced by neutrophils to fight infections. Here, we show that *N*-chlorination, induced by HOCl concentrations encountered at sites of inflammation, converts blood plasma proteins into chaperone-like holdases that protect other proteins from aggregation. This chaperone-like conversion was reversible by antioxidants and was abrogated by prior methylation of basic amino acids. Furthermore, reversible *N*-chlorination of basic amino acid side chains is the major factor that converts plasma proteins into efficient activators of immune cells. Finally, HOCl-modified serum albumin was found to act as a pro-survival molecule that protects neutrophils from cell death induced by highly immunogenic foreign antigens. We propose that activation and enhanced persistence of neutrophils mediated by HOCl-modified plasma proteins, resulting in the increased and prolonged generation of ROS, including HOCl, constitutes a potentially detrimental positive feedback loop that can only be attenuated through the reversible nature of the modification involved.
DOI: https://doi.org/10.7554/eLife.47395.001

## Introduction

Recruitment and activation of neutrophils at sites of infection is considered one of the principal mechanisms by which the human body protects itself against diseases. The killing strategy of neutrophils involves the ingestion of pathogens into the phagosome, accompanied by the production of a diverse set of so-called reactive oxygen species (ROS), including superoxide anions ($O_2^{\cdot-}$), hydrogen peroxide ($H_2O_2$) and hypochlorous acid (HOCl) in a process known as the respiratory burst (recently reviewed in *Mortaz et al., 2018* and *Thomas, 2017*).

HOCl, a major inflammatory ROS, is produced from hydrogen peroxide and chloride ions by the heme enzyme myeloperoxidase (MPO) (*Furtmüller et al., 1998*; *van Dalen et al., 1997*). The antimicrobial properties of HOCl are well documented and numerous reports have provided strong evidence for severe damage to bacterial components within the neutrophil phagosome (*Chapman et al., 2002*; *Rosen et al., 2002*; *Degrossoli et al., 2018*).

A particularly important target of HOCl are proteins. In proteins, HOCl exposure typically leads to side chain modification (*Clark et al., 1986*; *Beck-Speier et al., 1988*; *Heinecke, 1999*; *Domigan et al., 1995*; *Sivey et al., 2013*), fragmentation (*Vissers and Winterbourn, 1991*; *Hawkins and Davies, 1998*), misfolding/aggregation (*Hazell et al., 1994*) or intermolecular di-tyrosine cross-linking, a hallmark of HOCl-oxidized proteins (*Heinecke et al., 1993*). This often leads to their aggregation (*Müller et al., 2014*; *Chapman et al., 2003*; *Winter et al., 2008*). Consequently, various mechanisms to counter the accumulation of these misfolded proteins under HOCl stress exist in bacteria. These are mediated by chaperones that are activated by HOCl, the very same reactive species they protect against. One of the first protective proteins to be noticed was Hsp33, which gets reversibly activated by HOCl through oxidation of four critical cysteine residues (*Winter et al.,*

**\*For correspondence:**
lars.leichert@ruhr-uni-bochum.de

**Competing interests:** The authors declare that no competing interests exist.

*2008*). This is not surprising, as cysteine thiols together with methionine residues react rapidly with HOCl (*Storkey et al., 2014*; *Arnhold et al., 1991*). More recently, we found that *E. coli* RidA, a member of the highly conserved, but functionally diverse YjgF/YER057c/UK114 protein family, also undergoes HOCl-based conversion into a chaperone holdase. This chaperone was highly active as protector of proteins from HOCl-induced aggregation. However, in this case, instead of cysteine oxidation, *N*-chlorination of basic amino acids was the mechanism of its activation (*Müller et al., 2014*), although reactivity of HOCl with side chains of lysine and arginine is four to seven orders of magnitude lower than that of cysteine (*Pattison and Davies, 2001*). But much like cysteine oxidation, *N*-chlorination is a reversible oxidative modification that can be removed by cellular antioxidants such as ascorbate, glutathione or thioredoxin and thus can switch the holdase function of RidA off (*Müller et al., 2014*; *Chesney et al., 1996*; *Peskin and Winterbourn, 2001*).

Similar observations were made more recently with *E. coli* CnoX (YbbN) that, when activated via *N*-chlorination, binds to and prevents a variety of substrates from aggregation and being irreversible oxidized (*Goemans et al., 2018*).

While HOCl is highly bactericidal, generation of HOCl by immune cells is not without risk to the human body itself (reviewed in *Pechous, 2017*; *Thieblemont et al., 2016*; *Klebanoff, 2005*; *Davies, 2011*). During inflammatory processes, up to 30% of total cellular MPO is secreted by neutrophils into extracellular surroundings via degranulation, leakage during phagocytosis, or by association with NETs (*Parker et al., 2012*). Neutrophils, accumulated in the interstitial fluid of inflamed tissues, have been reported to generate HOCl at concentrations of up to 25–50 mM/hr (*Summers et al., 2008*). It is thus not surprising that HOCl can also drastically increase the activity of the extracellular chaperone $\alpha_2$-macroglobulin ($\alpha_2$M) in human blood plasma to counteract the aggregation of host proteins under hypochlorous acid stress (*Wyatt et al., 2014*). However, the underlying molecular mechanism of HOCl-mediated conversion of this plasma protein is still unclear.

And while neutrophils produce high amounts of HOCl, it does not accumulate at those levels, as it reacts instantly with diverse biological molecules including proteins (*Hawkins et al., 2003*), DNA (*Prütz, 1996*), cholesterol (*Carr et al., 1996*) and lipids (*Winterbourn et al., 1992*). But due to their high abundance in blood and interstitial fluid, human serum albumin (HSA) and other plasma proteins are considered the major targets of HOCl-mediated damage and as such constitute the main sink for HOCl in the vicinity of inflammation (*Shao et al., 2006*; *Peng et al., 2005*; *Colombo et al., 2017*; *Himmelfarb and McMonagle, 2001*; *Fogh-Andersen et al., 1995*). The resulting products of the reaction of plasma proteins with hypochlorous acid, known as advanced oxidation protein products (AOPPs), have, therefore, been employed as in vivo markers of chronic inflammation (*Witko-Sarsat et al., 1996*). Accumulation of AOPPs has been first discovered in patients with chronic kidney disease (*Witko-Sarsat et al., 1996*) and later also found in a variety of other inflammatory diseases, for example cardiovascular disease, neurodegenerative disorders, rheumatoid arthritis and some cancers (recently reviewed in *Aratani, 2018*).

To date, a number of studies have been carried out to elucidate the role of AOPPs in inflammatory processes (*Colombo et al., 2015*; *Liu et al., 2006*; *Descamps-Latscha et al., 2005*; *Witko-Sarsat et al., 2003*). Accumulating experimental evidence supports a critical contribution of AOPPs to the progression of inflammation (*Liu et al., 2006*). HOCl-modified HSA accumulates in inflammatory diseases and was found to act as proinflammatory mediator by increasing oxidative stress and inflammation through stimulation of leukocytes (*Witko-Sarsat et al., 2003*; *Gorudko, 2014*).

Based on these findings, we hypothesized that reversible *N*-chlorination in response to HOCl-stress could be the principal chemical modification contributing to the observed physiological properties of AOPPs. Furthermore, as we observed in previous studies that the HSA homologue bovine serum albumin (BSA) could be transformed into a chaperone-like holdase by HOCl-treatment (*Müller et al., 2014*), we speculated that *N*-chlorination serves as general rather than specific mechanism to transform certain proteins into a chaperone-like state.

Here, we study the effects of reversible *N*-chlorination on the function of human plasma proteins. We show that, upon *N*-chlorination, not only $\alpha_2$M, but all plasma fractions tested, exhibit chaperone-like activity and as such could prevent the HOCl-induced formation of protein aggregates at the site of inflammation. Moreover, exposure to HOCl at concentrations present in chronically inflamed tissues turned the majority of plasma proteins into efficient activators of neutrophil-like cells. Previous studies revealed that HOCl-treated HSA can stimulate leukocytes to produce more ROS during inflammation (*Gorudko, 2014*; *Witko-Sarsat et al., 1998*). Now we find that reversible *N*-

chlorination is the main chemical modification that mediates the activation of NADPH oxidase-dependent ROS generation by immune cells by AOPPs. Finally, we show that HOCl-modified HSA is a pro-survival factor for immune cells and protects neutrophils from cell death by highly immunogenic antigens.

Our data strongly suggest that in vivo reversible *N*-chlorination of human plasma proteins not only converts them into effective chaperone-like holdases but is also the principal mechanism that turns these proteins into mediators of the innate immune system.

## Results

### HOCl-treated human serum prevents protein aggregation

In previous experiments, we showed that the bacterial protein RidA is transformed into a competent holdase-type chaperone upon treatment with HOCl or *N*-chloramine (*Müller et al., 2014*). In control experiments, we discovered that bovine serum albumin also shows increased chaperone activity in response to HOCl (*Müller et al., 2014*). Since HOCl can be present in a physio-pathological context, most notably in the vicinity of inflammation, we wanted to test if human serum proteins can also be transformed into holdase-type chaperones through HOCl-treatment. Thus, we performed aggregation assays using chemically denatured citrate synthase and untreated and HOCl-treated human serum. Human serum was incubated with an estimated 10-fold molar excess of HOCl for 10 min at 30°C, a concentration that was sufficient to fully activate or markedly improve the chaperone function of RidA and BSA, respectively (*Müller et al., 2014*). When chemically denatured citrate synthase was diluted into denaturant-free buffer, it formed aggregates that can be monitored by increased light scattering of the solution (*Figure 1*). This aggregation of citrate synthase was not prevented by the addition of untreated human serum. However, when pre-incubated with a 10-fold molar excess of HOCl, serum significantly decreased aggregate formation. This suggested to us, that at least some serum proteins could be transformed to a holdase-type chaperone by *N*-chlorination in a mechanism similar to RidA. Because *N*-chlorination can be reduced by certain antioxidants, we used ascorbate to re-reduce HOCl-treated serum. Ascorbate is a mild antioxidant, which typically does not reduce native or HOCl-induced disulfide bonds. This ascorbate-treated serum lost its capability to bind denatured citrate synthase, suggesting an *N*-chlorination-based mechanism.

### Albumin, the major protein component of serum, shows HOCl-induced chaperone-like activity

The major protein component of human serum is human serum albumin (HSA). Previously, we showed that its bovine homologue bovine serum albumin (BSA) exhibits increased chaperone activity upon treatment with HOCl (*Müller et al., 2014*). We therefore suspected that HSA could be a major contributor to the observed decrease in protein aggregate formation in the presence of HOCl-treated serum. To test whether HSA gains chaperone function upon exposure to HOCl, HSA at a final concentration of 1 mM was treated with a 10-fold molar excess of HOCl for 10 min at 30°C. An HOCl concentration of at least 10 mM has been shown to be required for the generation of significant quantities of the so-called advanced oxidation protein products (AOPPs) in plasma, associated with a number of inflammatory diseases (*Witko-Sarsat et al., 1996*). The localized concentration of HOCl generated by accumulated neutrophils in the interstitium of chronically inflamed tissues, however, can be much higher and reach values of 25–50 mM/hr (*Summers et al., 2008*). To mimic a state of chronic inflammation, HSA was thus also incubated in the presence of a 50-fold molar excess of HOCl corresponding to concentrations ranging from 5 mM to 50 mM, depending on the protein concentration used. The typical interstitial concentration of HSA is assumed to be around 0.3 mM (*Smith and Staples, 1982*), bringing a 50-fold molar excess to 15 mM, a concentration that could reasonably be expected to be reached at confined sites of inflammation based on the propensity of neutrophils to generate HOCl (*Foote et al., 1983*). HSA treated in these ways significantly reduced the aggregation of citrate synthase, showing that HSA acts as a potent chaperone in serum upon modification by HOCl (*Figure 2a-d*). The chaperone activity of HSA was much higher upon exposure to a 50-fold molar excess of HOCl when compared to a 10-fold molar excess, suggesting a dose-dependent activation of the HSA chaperone function by HOCl. $H_2O_2$, another major oxidant generated by neutrophils, and a substrate for myeloperoxidase, did not convert HSA into a chaperone-like

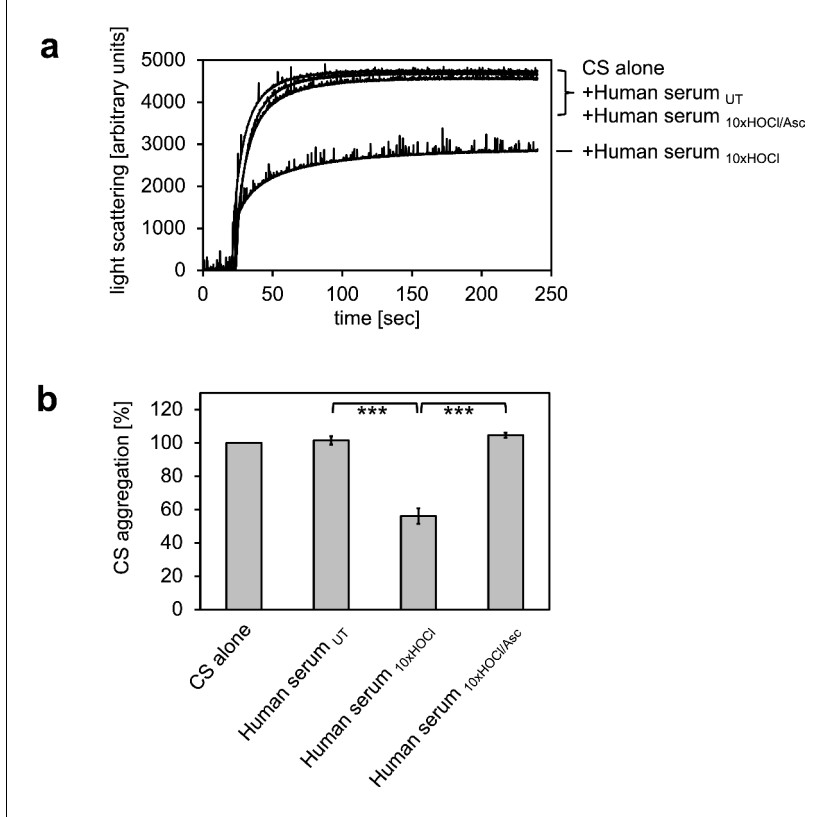

**Figure 1.** HOCl-treated human serum decreases protein aggregation. Human serum, when treated with a 10-fold molar excess of HOCl (Human serum $_{10xHOCl}$), significantly decreases aggregation of chemically denatured citrate synthase as measured by light scattering at 360 nm. Reduction of HOCl-treated human serum with a 50-fold molar excess of the antioxidant ascorbate (Human serum $_{10xHOCl/Asc}$) reverses this chaperone-like conversion of the serum. (a) A representative measurement of citrate synthase aggregation in the presence of untreated (Human serum $_{UT}$), HOCl-treated (Human serum $_{10xHOCl}$) and re-reduced (Human serum $_{10xHOCl/Asc}$) human serum is shown. (b) Data are represented as means and standard deviations from three independent aggregation assays. Student's t-test: ***p<0.001. Aggregation of citrate synthase in the absence of human serum was set to 100% and all the data are presented as percentage of this control. Labels of aggregation curves are written in the order of the final intensity of light scattering of the respective treatment.
DOI: https://doi.org/10.7554/eLife.47395.002

The following source data is available for figure 1:

**Source data 1.** Numerical light scattering data obtained during protein aggregation assays represented in *Figure 1a and b*.
DOI: https://doi.org/10.7554/eLife.47395.003

---

holdase (*Figure 2e,f*). The chaperone activity of HSA was not dependent on the substrate protein: HOCl-treated HSA also prevented the aggregation of IlvA (*Figure 2—figure supplement 1*). In contrast to bacterial RidA and Hsp33 (*Müller et al., 2014*; *Winter et al., 2008*), full activation of the chaperone function of HSA seems to require a higher oxidation/chlorination level of the protein.

## Chaperone-like conversion of serum albumin can be reversed by antioxidants

It is well known that exposure to high HOCl concentrations, such as those present at sites of chronic inflammation, can lead to plasma protein unfolding and the formation of carbonylated and di-tyrosine cross-linked protein aggregates that cannot return to the free, functional pool of proteins upon reduction by antioxidants (*Heinecke et al., 1993*). Such irreversibly misfolded plasma proteins could principally act as chaperones and bind to and prevent the aggregation of other unfolding substrates through hydrophobic interactions with the newly exposed hydrophobic protein surfaces.

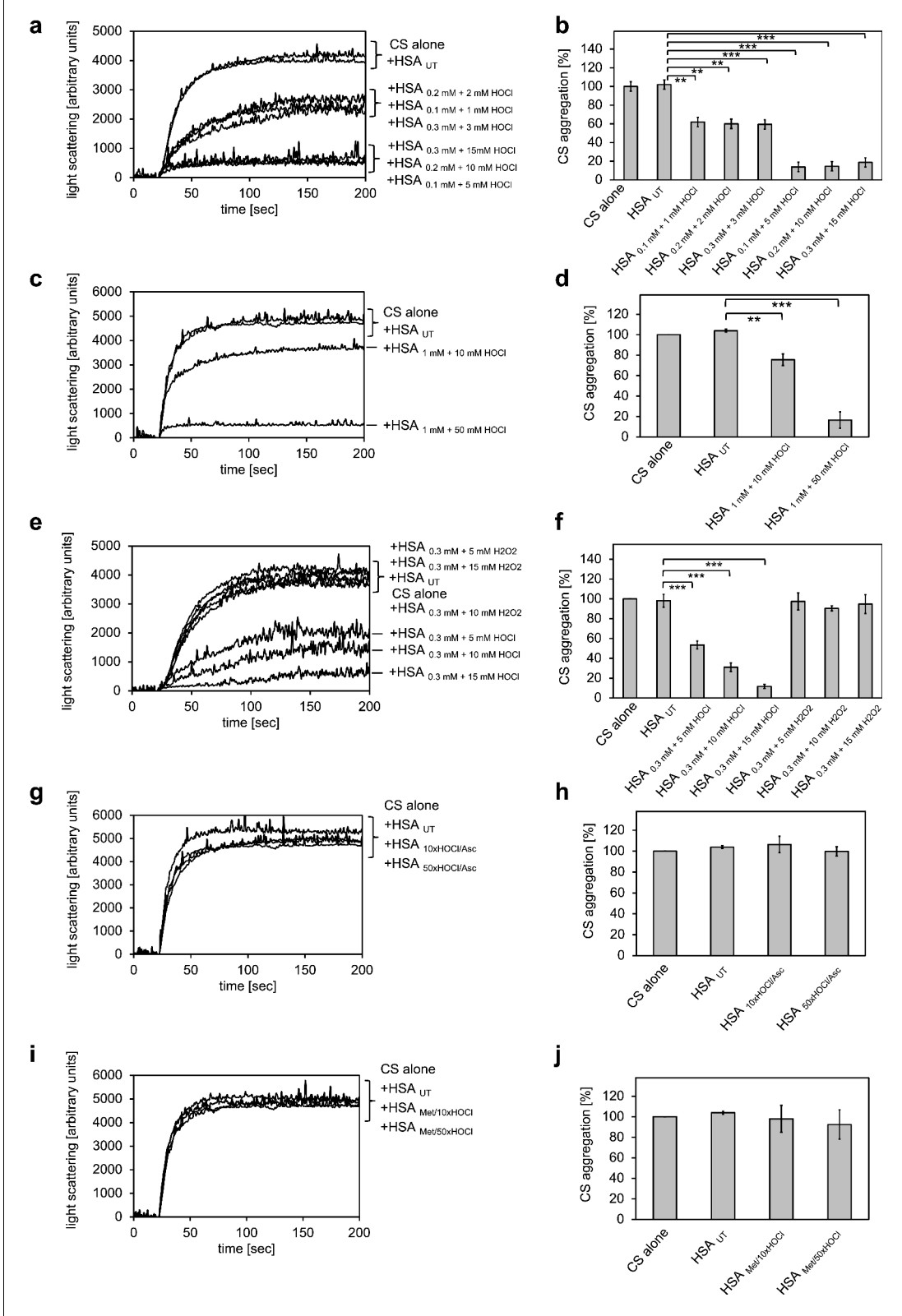

**Figure 2.** Conversion of serum albumin into a potent chaperone upon HOCl exposure is based on reversible *N*-chlorination of its basic amino acids. (a, b) Serum albumin in different concentrations, when treated with a 10- or 50-fold molar excess of HOCl (HSA $_{10xHOCl}$ and HSA $_{50xHOCl}$, respectively), significantly decreases aggregation of chemically denatured citrate synthase as measured by light scattering at 360 nm. (c, d) After testing several concentrations, we decided to perform further experiments with 1 mM HSA treated with a 10- and 50-fold excess of HOCl. (e, f) Treatment with $H_2O_2$
*Figure 2 continued on next page*

*Figure 2 continued*

does not mimic HOCl treatment. (**g, h**) Reduction of HOCl-treated HSA with a 50-fold molar excess of the antioxidant ascorbate (HSA $_{10xHOCl/Asc}$ and HSA $_{50xHOCl/Asc}$) switches off its chaperone activity. (**i, j**) Methylation of basic amino acid side chains prior to HOCl treatment (HSA $_{Met/10xHOCl}$ and HSA $_{Met/50xHOCl}$) abrogates the chaperone-like conversion of HSA. In **a, c, e, g, and i** representative measurements are shown. In **b, d, f, h, and j** data are represented as means and standard deviations from three independent experiments. Student's t-test: *p<0.05, **p<0.01, ***p<0.001. Aggregation of citrate synthase in the absence of HSA was set to 100% and all the data are presented as percentage of this control. Labels of aggregation curves are written in the order of the final intensity of light scattering of the respective treatment.

DOI: https://doi.org/10.7554/eLife.47395.004

The following source data and figure supplements are available for figure 2:

**Source data 1.** Numerical light scattering data obtained during protein aggregation assays represented in *Figure 2a and b*.
DOI: https://doi.org/10.7554/eLife.47395.013

**Source data 2.** Numerical light scattering data obtained during protein aggregation assays represented in *Figure 2c and d*.
DOI: https://doi.org/10.7554/eLife.47395.014

**Source data 3.** Numerical light scattering data obtained during protein aggregation assays represented in *Figure 2e and f*.
DOI: https://doi.org/10.7554/eLife.47395.015

**Source data 4.** Numerical light scattering data obtained during protein aggregation assays represented in *Figure 2g and h*.
DOI: https://doi.org/10.7554/eLife.47395.016

**Source data 5.** Numerical light scattering data obtained during protein aggregation assays represented in *Figure 2i and j*.
DOI: https://doi.org/10.7554/eLife.47395.017

**Figure supplement 1.** Serum albumin, when treated with a 50-fold molar excess of HOCl, significantly decreases aggregation of chemically denatured IlvA as measured by light scattering at 360 nm.
DOI: https://doi.org/10.7554/eLife.47395.005

**Figure supplement 1—source data 1.** Numerical light scattering data obtained during protein aggregation assays represented in *Figure 2—figure supplement 1*.
DOI: https://doi.org/10.7554/eLife.47395.006

**Figure supplement 2.** Serum albumin, when treated with a 50-fold molar excess of HOCl and then reduced with methionine, loses its propensity to decrease aggregation of chemically denatured citrate synthase.
DOI: https://doi.org/10.7554/eLife.47395.007

**Figure supplement 2—source data 1.** Numerical light scattering data obtained during protein aggregation assays represented in *Figure 2—figure supplement 2*.
DOI: https://doi.org/10.7554/eLife.47395.008

**Figure supplement 3.** Influence of reducing agents on the migration of HSA on reducing and non-reducing SDS PAGE gels.
DOI: https://doi.org/10.7554/eLife.47395.009

**Figure supplement 3—source data 1.** Original scans of gels represented in *Figure 2—figure supplement 3*.
DOI: https://doi.org/10.7554/eLife.47395.010

**Figure supplement 4.** Taurine N-chloramine, a model N-chloramine does not affect our light-scattering assay.
DOI: https://doi.org/10.7554/eLife.47395.011

**Figure supplement 4—source data 1.** Numerical light scattering data obtained during protein aggregation assays represented in *Figure 2—figure supplement 4*.
DOI: https://doi.org/10.7554/eLife.47395.012

However, based on the observation that the chaperone activity of serum albumin increased in a dose-dependent manner with the quantity of HOCl added, we asked whether the activation of HSA chaperone function upon treatment with a 10- or 50-fold molar excess of HOCl could be mediated by reversible *N*-chlorination instead. We have found this mechanism of action in the bacterial protein RidA (*Müller et al., 2014*). We thus exposed HOCl-treated HSA to a 50-fold molar excess of the antioxidant ascorbate, which specifically removes *N*-chlorination. Indeed, ascorbate rendered HSA $_{10xHOCl}$ and HSA $_{50xHOCl}$ unable to prevent citrate synthase aggregation when added at the same molar excess as the HOCl-treated protein samples (*Figure 2g,h*). Methionine, another known reducing agent for *N*-chloramines showed a similar effect (*Figure 2—figure supplement 2*). This result strongly supports the idea that HOCl-mediated activation of HSA chaperone function involves reversible chlorination of its side chain amines.

## Methylation of basic amino acid residues in serum albumin inhibits HOCl-induced activation of its chaperone function

Prompted by the observation that HOCl-mediated conversion of serum albumin into a potent chaperone could be reversed by ascorbate, we assumed an *N*-chlorination-based activation mechanism of its chaperone function. Because ascorbate could, in theory, also reduce certain transient thiol modifications such as sulfenylation or sulfenylchlorides, and because HSA contains a single non-disulfide-bound cysteine that could potentially undergo those modifications, we wanted to corroborate our hypothesis further. Thus, we blocked free amino groups of lysine and nitrogens in the guanidino-moiety of arginine residues in HSA via selective methylation. Exposure of methylated HSA to 10- or 50-fold molar excess of HOCl did not convert HSA into a chaperone, suggesting that activation of the HSA chaperone function indeed requires chlorination of its basic amino acids (*Figure 2i,j*). Analysis of HSA treated with HOCl and different reductants on reducing and non-reducing SDS gels did not reveal formation of an HSA dimer linked by an intermolecular disulfide bond nor did it show evidence that ascorbate or methionine was able to reduce HSA's intrinsic disulfides, whereas DTT reduction led to an observable migration shift (*Figure 2—figure supplement 3*).

## Decreased amino group content of HSA upon HOCl treatment is accompanied by an increased overall hydrophobicity of the protein

Our combined data strongly suggested that the chaperone function of HSA is activated by chlorination of its basic amino acids. HSA in its secreted form possesses 59 primary amines in the form of sidechains of lysine, which could act as targets for chlorination by HOCl. There are an additional 24 arginine residues, which contain a guanidino group, as well as 16 histidines and one tryptophane, which contain nitrogen in an aromatic ring system, and one amino group at the N-terminus (*Temple et al., 2006*), which could potentially be chlorinated as well. To exclude that simply the presence of these *N*-chlorinated species affects our light-scattering assay, we performed a chaperone assay in the presence of a comparable amount of taurine *N*-chloramine as a control. This had no effect (*Figure 2—figure supplement 4*). To investigate the extent to which HOCl decreases the total amount of accessible, non-modified amino groups in HSA, we analyzed the free amino group content before and after HOCl-treatment using fluorescamine (*Udenfriend et al., 1972*). Evidently, exposure to HOCl resulted in some loss of free amino groups. Amino group content of HSA decreased by approximately 10% after treatment with a 10-fold molar excess of HOCl (HSA $_{10xHOCl}$) and by 40% upon exposure to a 50-fold molar excess of HOCl (HSA $_{50xHOCl}$) (*Figure 3a,b*). Reduction of both chlorinated HSA samples with ascorbate resulted in a full recovery of accessible amino groups with concomitant loss of chaperone activity (*Figure 2g,h*). Activation of HSA chaperone function by HOCl thus coincides with a decrease in free amino group content, providing further evidence for an *N*-chlorination based-mechanism. A direct determination of *N*-chloramines showed that content of *N*-chloramines increases with excess of HOCl in accordance with the decrease of free amino groups. This increase was reversible by ascorbate, DTT and methionine and virtually no chloramines were detected in samples treated with these reductants (*Figure 3c,d*, *Figure 3—figure supplement 1*).

We argued that the reduction of positive charges on the protein's surface through HOCl-induced *N*-chlorination should lead to an increase in surface hydrophobicity, thus allowing high-affinity binding to unfolded proteins.

To detect changes in HSA's surface hydrophobicity upon HOCl treatment, we used the uncharged hydrophobic dye Nile red and measured its fluorescence upon addition to 25 μM native HSA or 25 μM HSA $_{50xHOCl}$ (*Sackett and Wolff, 1987*). Absolute fluorescence of 1.6 μM Nile red was higher for HSA $_{50xHOCl}$ when compared with untreated HSA with a maximum at 616.5 nm (621.5 nm for native HSA) showing a blue shift in emission maximum consistent with a decreased polarity of the protein solution (*Figure 3e*). The concentration of native HSA and HSA $_{50xHOCl}$, at which the proteins have been half-saturated with dye were calculated (*Figure 3f*). This concentration was significantly higher for untreated HSA compared to HSA $_{50xHOCl}$, pointing toward an increased hydrophobicity of HSA $_{50xHOCl}$.

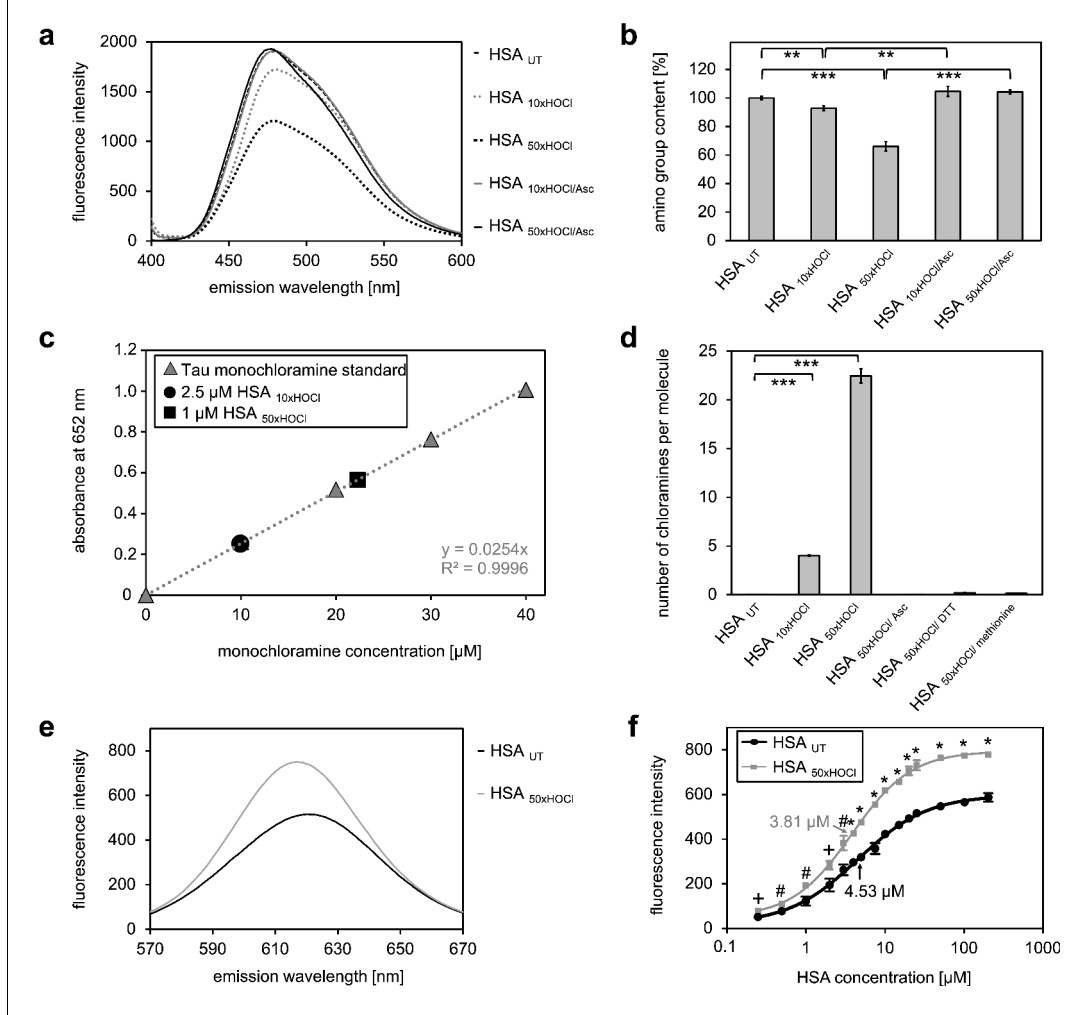

**Figure 3.** *N*-chlorination of serum albumin decreases accessible amino group content and increases chloramine content and surface hydrophobicity. (**a**, **b**) Amino group content of variously treated HSA was analyzed using fluorescamine. Treatment of HSA with HOCl resulted in a dose-dependent loss of free amino groups. Reduction of chlorinated HSA with ascorbate fully restored the free amino group content. (**c**) Chloramine content of HSA treated with 10- or 50-fold molar excess of HOCl. Chloramines were determined in 2.5 µM HSA treated with a 10-fold excess of HOCl and 1 µM HSA treated with a 50-fold excess and compared to a standard curve generated with known quantities of taurine N-chloramine. (**d**) Quantified chloramine content in HSA treated with a 10- and 50-fold excess of HOCl. (**e**) Fluorescence of 1.6 µM Nile red in the presence of native HSA or HSA, that has been treated with a 50-fold molar excess of HOCl. An increased absolute fluorescence and a shift in maximum emission wavelength (from 621.5 nm to 616.5 nm) can be observed for HSA $_{50xHOCl}$. (**f**) Absolute fluorescence of Nile red measured at the emission maximum at 621.5 nm and 616.5 nm, respectively, was plotted against the corresponding HSA $_{UT}$ and HSA $_{50xHOCl}$ concentrations, respectively. The average concentrations, at which HSA $_{UT}$ and HSA $_{50xHOCl}$ have been half-saturated with Nile red are marked by arrows. Means and standard deviations in **b**, **d**, and **f** are based on three independent experiments. (**b and d**) Student's t-test: *p<0.05, **p<0.01, ***p<0.001. (**f**) Student's t-test:+p < 0.05, #p<0.01, *p<0.001. For **a**, **c**, and **d** representative measurements are shown.

DOI: https://doi.org/10.7554/eLife.47395.018

The following source data and figure supplements are available for figure 3:

**Source data 1.** Numerical fluorescence spectroscopy data obtained during determination of free amino groups represented in *Figure 3a and b*.
DOI: https://doi.org/10.7554/eLife.47395.021
**Source data 2.** Numerical fluorescence spectroscopy data obtained during determination of protein chloramines represented in *Figure 3c*.
DOI: https://doi.org/10.7554/eLife.47395.022
**Source data 3.** Numerical fluorescence spectroscopy data obtained during determination of protein chloramines represented in *Figure 3d*.
DOI: https://doi.org/10.7554/eLife.47395.023
**Source data 4.** Numerical fluorescence spectroscopy data obtained during determination of protein hydrophobicity represented in *Figure 3e*.
DOI: https://doi.org/10.7554/eLife.47395.024
**Source data 5.** Numerical fluorescence spectroscopy intensity data obtained during determination of protein hydrophobicity represented in *Figure 3f*.

*Figure 3 continued on next page*

*Figure 3 continued*

DOI: https://doi.org/10.7554/eLife.47395.025

**Figure supplement 1.** Determination of the *N*-chloramine content of HOCl-treated HSA.

DOI: https://doi.org/10.7554/eLife.47395.019

**Figure supplement 1—source data 1.** Numerical fluorescence spectroscopy data obtained during determination of protein chloramines represented in *Figure 3—figure supplement 1*.

DOI: https://doi.org/10.7554/eLife.47395.020

## Activation of neutrophil-like cells by HOCl-treated serum albumin is based on reversible *N*-chlorination

Several lines of evidence point toward a key role of HOCl-modified serum albumin in the progression of chronic inflammation, a hallmark of various degenerative diseases (*Witko-Sarsat et al., 2003*; *Gorudko, 2014*). Upon exposure to high doses of HOCl, as those present in chronically inflamed tissues, modified HSA was found to induce neutrophil NADPH oxidase activation reflected by an increased generation of reactive oxygen species and accompanied by degranulation (*Gorudko, 2014*).

Pathophysiological concentrations of HOCl are able to induce different modifications on plasma proteins including carbonylation, *N*-chlorination, cysteine and methionine oxidation or inter- and intramolecular di-tyrosine cross-linking (*Clark et al., 1986*; *Beck-Speier et al., 1988*; *Domigan et al., 1995*; *Heinecke et al., 1993*), with most of them being considered irreversible. So far, it has not been elucidated which HOCl-induced modification is sufficient to convert HSA into a potent activator of leukocytes. We thus wondered whether this functional conversion of HSA might be specifically mediated by *N*-chlorination and thus reflects a reversible process, similar to the activation of its chaperone function in response to HOCl stress.

To test this, we analyzed the effect of HOCl-treated HSA before and after reduction by antioxidants on the activity of the phagocytic NADPH oxidase. For this purpose, we chose the human myeloid cell line PLB-985 that acquires a neutrophil-like phenotype upon differentiation with DMSO and IFNγ (*Pivot-Pajot et al., 2010*; *Tlili et al., 2011*).

Generation of oxidants by the NADPH oxidase was assessed by lucigenin, a well characterized and frequently used chemiluminescence probe, which reacts with superoxide anion radicals and hydrogen peroxide to form a light-emitting species (*Aasen et al., 1987*; *Maskiewicz et al., 1979*). Differentiated PLB-985 cells, when incubated in buffer in the absence of any activating agents, showed a low basal level of superoxide generation resulting from IFNγ-mediated enhancement of the NADPH oxidase activity during the differentiation period (*Ellison et al., 2015*) (*Figure 4a*). In line with expectations, treatment of the cells with phorbol-12-myristate-13-acetate (PMA), a known activator of neutrophil NADPH oxidase, led to a drastic increase in superoxide production and was thus used as positive control (*Wolfson et al., 1985*) (*Figure 4a,c*). Addition of untreated HSA significantly increased the generation of superoxide by >20% compared to the mock control. No further enhancement of the NADPH oxidase activity was observed with HSA after exposure to a 10-fold molar excess of HOCl, indicating that the chosen HOCl concentration might have been too low for the conversion of HSA into a potent activator of neutrophil-like cells. In contrast, cells subjected to HSA that has been treated with a 50-fold molar excess of HOCl exhibited a substantial increase in superoxide production within the first ~25 min of incubation. This increase was observed both for preparations of 1 mM HSA treated with 50 mM HOCl and 0.3 mM HSA treated with 15 mM HOCl (*Figure 4—figure supplement 1*). In comparison to PMA, NADPH oxidase activation by HSA $_{50xHOCl}$ proceeded at a lower rate and was less sustained, returning to basal level 40 min after the addition of HSA $_{50xHOCl}$ to the cells.

During our experiments, we observed that *N*-chlorinated HSA can directly react with the fluorescent dye, 2′, 7′-dichlorodihydrofluorescein diacetate (H$_2$DCF-DA), which is the most widely used probe for measuring intracellular ROS production (*Chen et al., 2010*), eliciting a positive signal in the absence of cells (*Figure 4—figure supplement 2*). To confirm that the observed chemiluminescent signal emitted by lucigenin derives from HSA $_{50xHOCl}$-induced activation of the phagocytic NADPH-oxidase rather than from a similar direct reaction of lucigenin with HSA $_{50xHOCl}$ -derived chloramines, we incubated lucigenin with various agents, including taurine-*N*-chloramine and

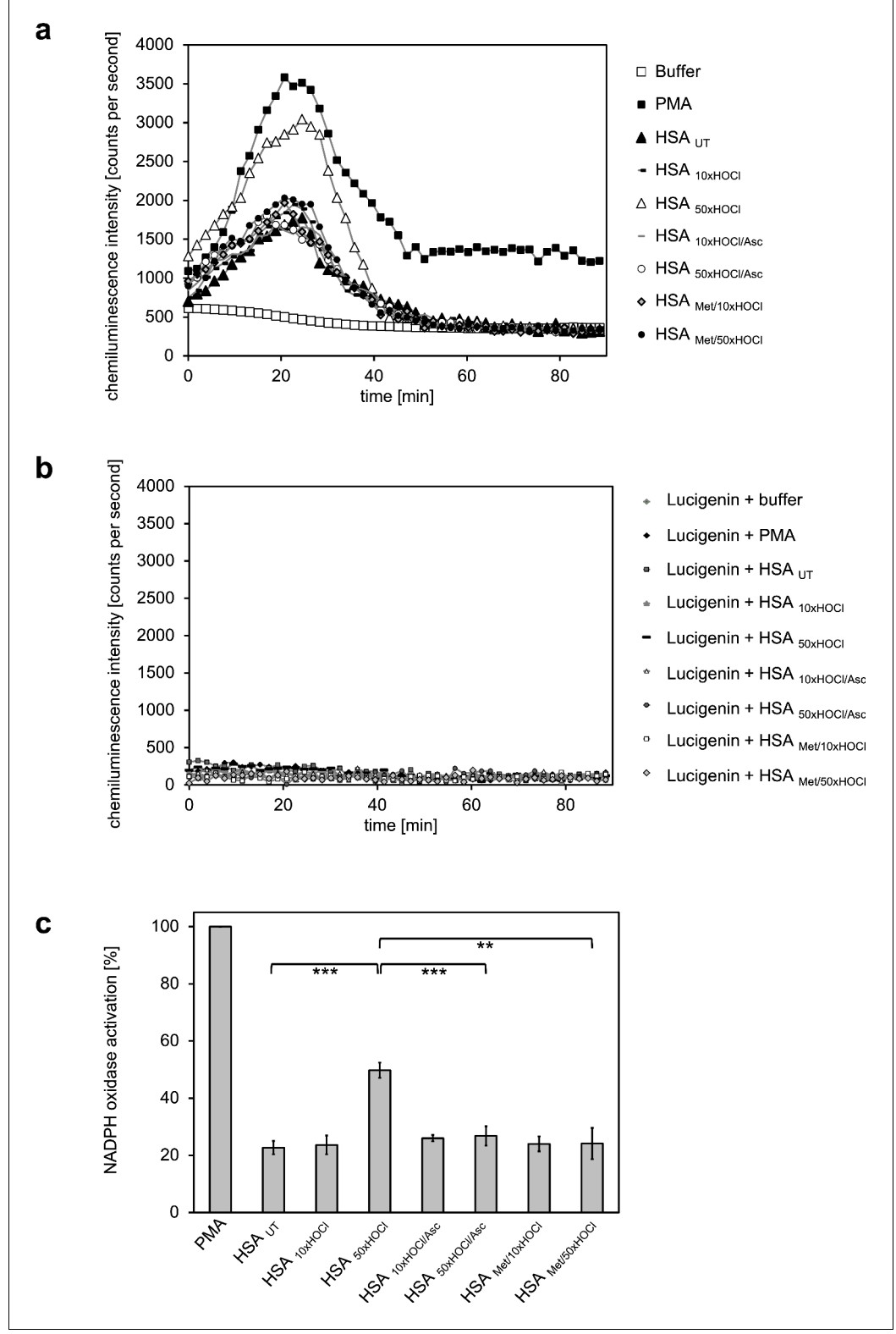

**Figure 4.** Activation of neutrophil-like cells by HOCl-treated serum albumin is mediated by reversible *N*-chlorination. Treatment with a 50-fold molar excess of HOCl (HSA $_{50xHOCl}$) converted HSA into an efficient inducer of the neutrophil respiratory burst, reflected by the increased production and release of oxidants that induce lucigenin chemiluminescence. This activating function of HSA $_{50xHOCl}$ could be reversed by reduction with the antioxidant ascorbate (HSA $_{50xHOCl/Asc}$) and was abrogated by methylation of its basic amino acid side chains prior

*Figure 4 continued on next page*

*Figure 4 continued*

to HOCl exposure (HSA $_{Met/50xHOCl}$). (a) Extracellular oxidant production by neutrophil NADPH oxidase was measured in one- to two-minutes intervals over 90 min at 37°C using lucigenin-enhanced chemiluminescence. Phorbol 12-myristate 13-acetate (PMA; final concentration, 0.2 μM), untreated and variously treated HSA samples (final concentration, 3 mg · mL$^{-1}$) from the previous citrate synthase aggregation assays (see above) or PBS buffer (basal oxidant production) were added to (a) differentiated PLB-985 cells in PBS buffer or (b) cell-free PBS puffer containing 400 μM lucigenin immediately prior to chemiluminescence measurement. (c) Results shown in a are expressed as integrated total counts (means and standard deviations of three independent measurements) higher than buffer control. Student's t-test: **p<0.01, ***p<0.001. PMA-induced activation of NADPH-oxidase was set to 100%.

DOI: https://doi.org/10.7554/eLife.47395.026

The following source data and figure supplements are available for figure 4:

**Source data 1.** Numerical chemiluminescence plate reader data represented in *Figure 4a and c*.

DOI: https://doi.org/10.7554/eLife.47395.033

**Source data 2.** Numerical chemiluminescence plate reader data represented in *Figure 4b*.

DOI: https://doi.org/10.7554/eLife.47395.034

**Figure supplement 1.** Activation of neutrophil-like cells by HOCl-treated serum albumin.

DOI: https://doi.org/10.7554/eLife.47395.027

**Figure supplement 1—source data 1.** Numerical chemiluminescence plate reader data represented in *Figure 4—figure supplement 1*.

DOI: https://doi.org/10.7554/eLife.47395.028

**Figure supplement 2.** N-chlorinated serum albumin converts 7'-dichlorodihydrofluorescein diacetate (H$_2$DCF-DA) to the fluorescent 2', 7'-dichlorofluorescein (DCF).

DOI: https://doi.org/10.7554/eLife.47395.029

**Figure supplement 1—source data 1.** Numerical chemiluminescence plate reader data represented in *Figure 4—figure supplement 2*.

DOI: https://doi.org/10.7554/eLife.47395.030

**Figure supplement 3.** Taurine N-chloride and hydrogen peroxide-treated HSA do not induce the respiratory burst in neutrophil-like cells.

DOI: https://doi.org/10.7554/eLife.47395.031

**Figure supplement 3—source data 1.** Numerical chemiluminescence plate reader data represented in *Figure 4—figure supplement 3*.

DOI: https://doi.org/10.7554/eLife.47395.032

hydrogen peroxide-treated HSA, in the presence and absence of cells and no significant chemiluminescence was observed (*Figure 4b*, *Figure 4—figure supplement 3*).

Importantly, when reduced by ascorbate, HSA $_{50xHOCl}$ lost its stimulatory effect, showing that the functional conversion of HSA to an efficient activator of neutrophil-like cells upon HOCl exposure reflects a reversible process. Since ascorbate specifically reduces *N*-chloramines without affecting many other HOCl-induced modifications, these results strongly support an *N*-chlorination-based mechanism. The latter finding was further corroborated by methylation of the basic amino acids of HSA prior to HOCl treatment, which completely prevented HOCl-treated HSA from activating the neutrophil oxidative burst.

*N*-chlorinated HSA, formed upon exposure to high doses of HOCl, can be thus considered as an inflammatory response modulator that contributes to the activation of neutrophils at sites of inflammation (*Figure 4c*).

## All plasma fractions tested exhibit chaperone activity upon HOCl treatment

Our finding that the major protein in human plasma, serum albumin can be converted into a potent chaperone upon HOCl-mediated *N*-chlorination, prompted us to ask whether also other plasma protein fractions exhibit similar chaperone activity upon treatment with HOCl.

We thus tested the γ-globulin fraction, the Cohn fraction IV (comprising α- and β-globulins), and specifically α$_2$-macroglobulin, an important protease inhibitor in human plasma, for chaperone activity upon exposure to various doses of HOCl.

Aside from its known interaction with proteases, native $\alpha_2$-macroglobulin ($\alpha_2$M) acts as an extracellular chaperone that binds to and prevents the accumulation of misfolded proteins, particularly during the innate immune response. Exposure to HOCl was found to further improve the chaperone function of $\alpha_2$M, but the mechanism remained unclear (*Wyatt et al., 2014*).

To test whether the previously observed effect is due to *N*-chlorination, we treated $\alpha_2$M with a 10- and 50-fold molar excess of HOCl (corresponding to 0.3 mM and 1.5 mM HOCl, respectively), followed by the addition of the antioxidant ascorbate for re-reduction. In line with expectations, chaperone activity of $\alpha_2$M increased with the amount of HOCl added (*Figure 5a,b*). Treatment with ascorbate, as well as methylation of basic amino acid residues prior to HOCl exposure fully inhibited the chaperone activation of HOCl-modified $\alpha_2$M. These results strongly support the notion that HOCl-induced *N*-chlorination of basic amino acid side chains also accounts for the increased chaperone activity of $\alpha_2$M.

Intriguingly, we also observed similar chaperone-like conversion upon HOCl treatment for Cohn fraction IV and the $\gamma$-globulin fraction (*Figure 5c–f*). In both cases, however, efficient activation of the chaperone activity required higher HOCl concentrations. While exposure of Cohn fraction IV to an estimated 10-fold molar excess of HOCl (corresponding to 2 mM HOCl) had only little effect on citrate synthase aggregation, treatment with a 50-fold molar excess of HOCl strongly activated chaperone-like properties (*Figure 5c,d*). This chaperone activity was markedly reduced, but not completely inhibited, after reduction with ascorbate. Likewise, methylation of amine side chains prior to HOCl exposure did not fully abrogate the chaperone-like conversion of this protein fraction.

Similarly, activation of the chaperone function of the $\gamma$-globulin fraction occurred at higher HOCl concentrations ranging from 4.3 mM to 13 mM (corresponding to an estimated 50- to 150-fold molar excess of HOCl). Strikingly, while chaperone activity increased with the amount of HOCl added, reversibility of the chaperone function by ascorbate decreased (*Figure 5e,f*).

These results suggest that at least some proteins in both plasma fractions tested are transformed to chaperones upon modification by HOCl. Efficient activation of their chaperone function is strongly, but, unlike HSA, not exclusively linked to *N*-chlorination, suggesting some other HOCl-induced modifications that cannot be removed by ascorbate in these plasma proteins.

## HOCl-induced *N*-chlorination converts the majority of plasma proteins into activators of neutrophil-like cells

Ours and others findings indicate that *N*-chlorinated serum albumin is a key factor for the stimulation of leukocytes at sites of inflammation. To test if also other plasma proteins exert this function upon HOCl exposure, we analyzed the NADPH oxidase-dependent generation of superoxide by differentiated PLB-985 cells in the presence of various treated plasma protein fractions, as described above.

Addition of $\alpha_2$-macroglobulin, treated with a 50-fold molar excess of HOCl (i.e. 1.5 mM HOCl), did not enhance superoxide production (*Figure 6a,b*). In contrast, exposure to the same molar excess of HOCl, corresponding to 10 mM HOCl in this case, converted at least some proteins of Cohn fraction IV into highly efficient stimulators of the neutrophil NADPH oxidase (*Figure 6c,d*). Analysis of the kinetics showed that activation by this HOCl-treated protein fraction occurred at a similar rate as the activation by PMA. Such a stimulatory effect was also observed for the $\gamma$-globulin fraction upon treatment with an estimated 150-fold molar excess of HOCl (i.e. 13 mM). The minimum HOCl concentration required for the activating function of these plasma proteins was thus 10 mM and conformed to the range of HOCl concentrations expected under inflammatory conditions. Importantly, the activating effect of both HOCl-treated plasma fractions was completely abolished upon reduction with ascorbate or by prior methylation of their amino groups, strongly suggesting that *N*-chlorination is the responsible mechanism for the functional switch of these proteins into efficient activators of neutrophil-like cells as well.

These findings demonstrate that HOCl-mediated *N*-chlorination constitutes a key mechanism to increase the immunogenicity of plasma proteins under inflammatory conditions. Upon modification, not only serum albumin, but the majority of the plasma fractions tested, form a feed-forward inflammatory loop to amplify and sustain inflammatory responses which can lead to accelerated pathogen clearance but could also contribute to chronic inflammation.

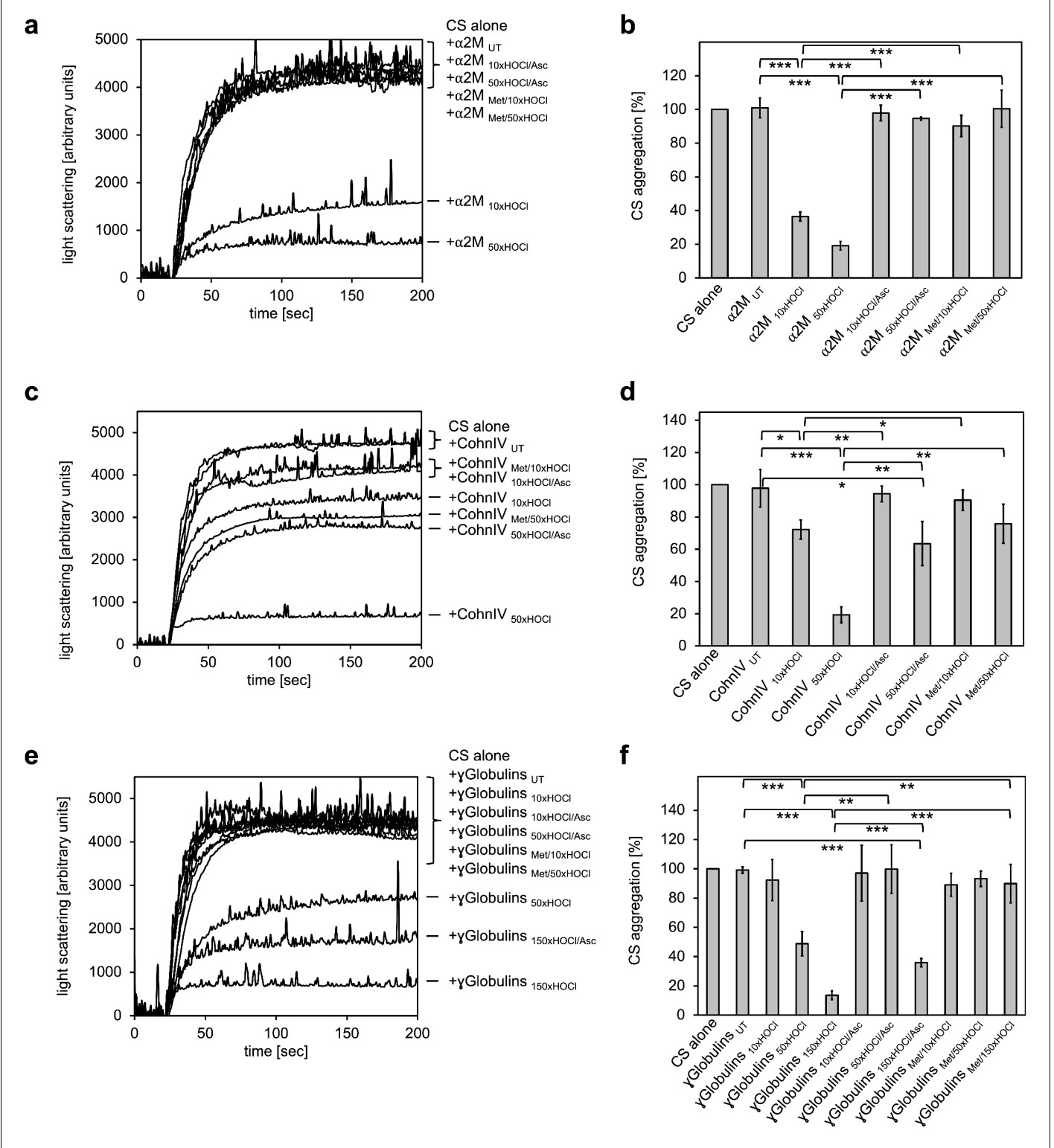

**Figure 5.** All plasma protein fractions tested exhibit reversible chaperone activity upon modification by HOCl. $\alpha_2$-Macroglobulin (**a, b**), Cohn fraction IV (**c, d**) and the $\gamma$-globulin fraction (**e, f**) were analyzed for chaperone activity in a citrate synthase aggregation assay upon treatment with various doses of HOCl. Each plasma protein fraction, when treated with a 10-, 50- or 150-fold molar excess of HOCl, significantly decreased aggregation of chemically denatured citrate synthase as measured by light scattering at 360 nm. Exposure of the various HOCl-treated plasma proteins to the reductant ascorbate significantly decreased or completely inhibited their chaperone function. Methylation of basic amino acid residues prior to HOCl treatment mostly prevented chaperone-like conversion of the plasma proteins. In **a, c** and **e** representative measurements are shown. In **b, d** and **f** data

*Figure 5 continued on next page*

*Figure 5 continued*

are depicting means and standard deviations from three independent experiments. Student's t-test: *p<0.05, **p<0.01, ***p<0.001. Aggregation of citrate synthase in the absence of any plasma protein fraction was set to 100% and all the data are presented as percentage of this control. Labels of aggregation curves are written in the order of the final intensity of light scattering of the respective treatment.

DOI: https://doi.org/10.7554/eLife.47395.035

The following source data is available for figure 5:

**Source data 1.** Numerical light scattering data obtained during protein aggregation assays represented in *Figure 5a and b*.
DOI: https://doi.org/10.7554/eLife.47395.036

**Source data 2.** Numerical light scattering data obtained during protein aggregation assays represented in *Figure 5c and d*.
DOI: https://doi.org/10.7554/eLife.47395.037

**Source data 3.** Numerical light scattering data obtained during protein aggregation assays represented in *Figure 5e and f*.
DOI: https://doi.org/10.7554/eLife.47395.038

## Activation of NADPH oxidase by *N*-chlorinated serum albumin and immunoglobulin G occurs predominantly via PI3K-dependent signaling pathways

Activity of the NADPH oxidase complex of neutrophils is regulated by several signaling pathways downstream of cell surface receptors. Central amongst these are PLC/PKC- (phospholipase C/protein kinase C) and PI3K- (phosphoinositide 3-kinase)-dependent pathways, blockade of which severely lowers NADPH oxidase activation by various stimuli such as chemotactic peptides, opsonized particles or phorbol esters (*Combadière et al., 1993*; *Bissonnette et al., 2008*; *Simonsen and Stenmark, 2001*; *Perisic et al., 2004*).

To identify the predominant signaling mechanism through which the *N*-chlorinated serum albumin and the major component of the γ-globulin fraction, immunoglobulin G, activate the neutrophil respiratory burst, the effect of various inhibitors on the HSA $_{50xHOCl}$ and IgG $_{150xHOCl}$-induced ROS generation was tested. PMA, a direct activator of conventional PKC isoforms such as PKCα and PKCβ was used as a control for PKC-dependent activation of the NADPH oxidase (*Wolfson et al., 1985*).

It is worth noting, that all cell suspensions contained DMSO at a final concentration of 1%. As a consequence, NADPH oxidase activation by PMA, HSA $_{50xHOCl}$ or IgG $_{150xHOCl}$ proceeded at a lower rate compared to the previously described experiments (see *Figures 4* and *6*), peaking at 35–50 min after the addition of these agents to the cells.

Pretreatment of neutrophil-like cells with 10 µM diphenyleneiodonium (DPI), a direct inhibitor of the NADPH oxidase complex, for 30 min prior to stimulation with PMA, HSA $_{50xHOCl}$ or IgG $_{150xHOCl}$ fully inhibited ROS generation confirming that the ROS production induced by these stimulatory agents was NADPH-oxidase dependent (*Figure 7*). Addition of catalase (CAT), superoxide dismutase (SOD) or both attenuated lucigenin chemiluminescence as well, further implicating NADPH oxidase (*Figure 7—figure supplement 1*). In line with expectations, presence of the PI3K inhibitor wortmannin (100 nM) had only little effect on the PMA-mediated activation of the NADPH oxidase (*Figure 7c,d*). In contrast, wortmannin strongly attenuated the HSA $_{50xHOCl}$- and IgG $_{150xHOCl}$-induced ROS generation by the immune cells, suggesting a PI3K-dependent mechanism of NADPH oxidase activation (*Figure 7a,b,d*). Along this line, pretreatment with 200 nM Gö 6983, that selectively inhibits several PKC isozyme families, including the classical PKCα and PKCβII, the novel PKCδ and the atypical PKCζ, strongly inhibited NADPH oxidase activation by PMA, but not by HSA $_{50xHOCl}$ and IgG $_{150xHOCl}$.

These results suggest that PI3K is the key signaling component in the pathway that leads to HSA $_{50xHOCl}$- and IgG $_{150xHOCl}$-dependent activation of the neutrophil NADPH oxidase.

## HOCl-treated serum albumin promotes survival of neutrophil-like cells in the presence of the foreign protein antigen Ag85B, but not in the presence of staurosporine

As shown previously, activation of the NADPH oxidase by HOCl-modified serum albumin involves the action of PI3K. Since PI3K is a key component of the well-documented, anti-apoptotic PI3K/Akt signaling pathway (*Martelli et al., 2007*), it was tempting to speculate that HOCl-modified HSA

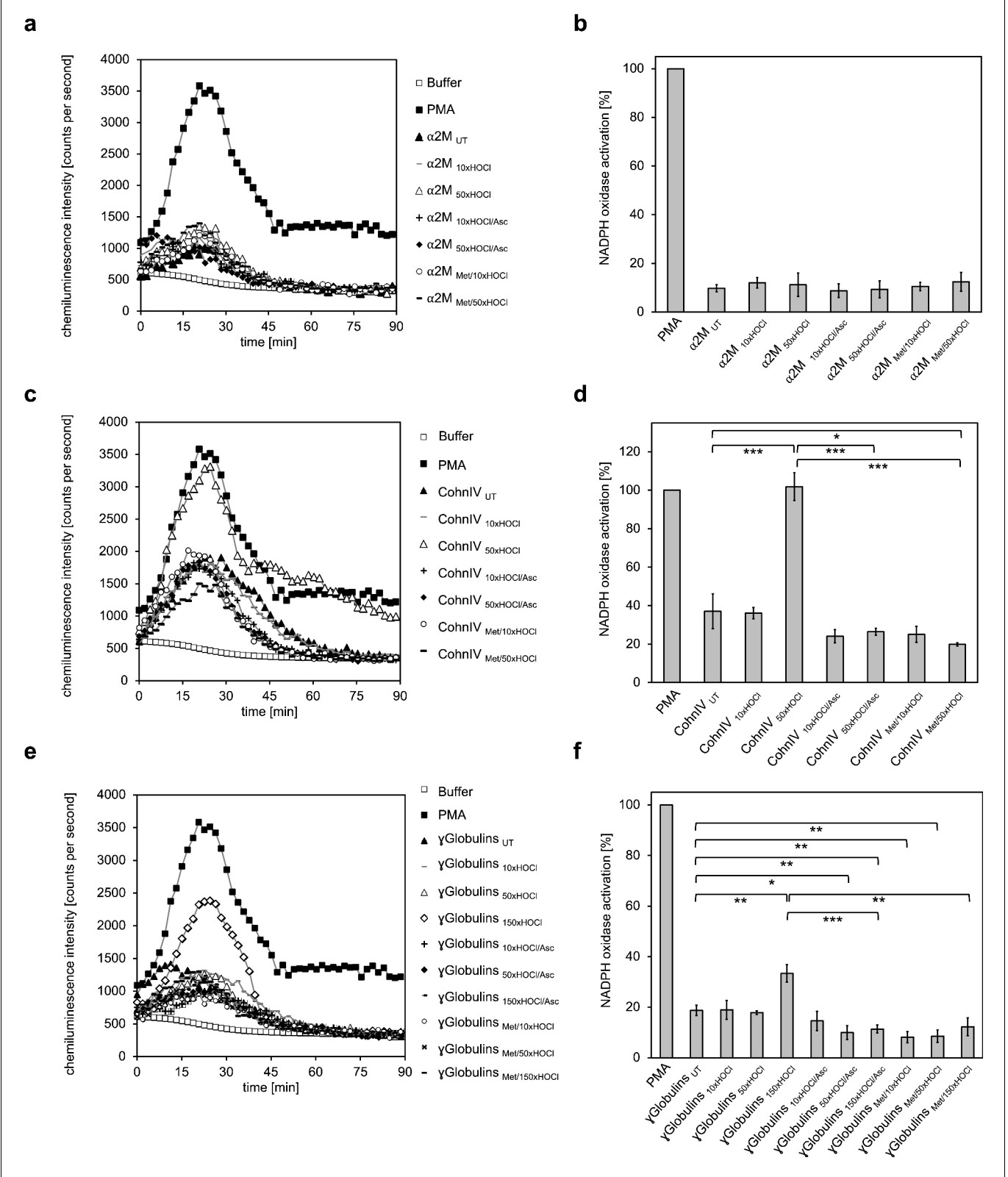

**Figure 6.** The majority of human plasma proteins stimulate neutrophil respiratory burst upon *N*-chlorination by HOCl. The effect of HOCl-treated $\alpha_2$-macroglobulin ($\alpha_2$M) (**a, b**), Cohn fraction IV (**c, d**) and the γ-globulin fraction (**e, f**) on the activity of the neutrophil NADPH oxidase was investigated. $\alpha_2$M, when treated with various doses of HOCl, had no influence on ROS generation by the NADPH oxidase (**a, b**). Treatment with a 50- or 150-fold molar excess of HOCl converted at least some proteins of Cohn fraction IV (CohnIV $_{50xHOCl}$) and the γ-globulin fraction (γGlobulins $_{150xHOCl}$),

*Figure 6 continued on next page*

*Figure 6 continued*

respectively, into efficient inducers of the neutrophil respiratory burst, reflected by the increased production and release of oxidants that induce lucigenin chemiluminescence (c-f). The activating function of CohnIV $_{50xHOCl}$ and γGlobulins $_{150xHOCl}$ could be reversed by treatment with the reductant ascorbate (CohnIV $_{50xHOCl/Asc}$ and γGlobulins $_{150xHOCl/Asc}$) and was abrogated by methylation of basic amine side chains prior to HOCl exposure (CohnIV $_{Met/50xHOCl}$ and γGlobulins $_{Met/150xHOCl}$). (a, c, e) Extracellular oxidant production by neutrophil NADPH oxidase was measured in one- to two-minutes intervals over 90 min at 37˚C using lucigenin-enhanced chemiluminescence. Phorbol 12-myristate 13-acetate (PMA; final concentration (fc), 0.2 µM), untreated and the variously treated plasma fraction samples (fc, 2 mg · mL$^{-1}$ for α$_2$-macroglobulin and 3 mg · mL$^{-1}$ for Cohn fraction IV and the γ-globulin fraction) from the previous citrate synthase aggregation assays (see above) or PBS buffer (basal oxidant production) were added to differentiated PLB-985 cells in PBS buffer containing 400 µM lucigenin immediately prior to chemiluminescence measurement. (b, d, f) Results are expressed as integrated total counts (means and standard deviations of three independent measurements) higher than buffer control. Student's t-test: *p<0.05, **p<0.01, ***p<0.001. PMA- induced activation of NADPH-oxidase was set to 100%.

DOI: https://doi.org/10.7554/eLife.47395.039

The following source data is available for figure 6:

**Source data 1.** Numerical chemiluminescence plate reader data represented in *Figure 6a and b*.
DOI: https://doi.org/10.7554/eLife.47395.040
**Source data 2.** Numerical chemiluminescence plate reader data represented in *Figure 6c and d*.
DOI: https://doi.org/10.7554/eLife.47395.041
**Source data 3.** Numerical chemiluminescence plate reader data represented in *Figure 6e and f*.
DOI: https://doi.org/10.7554/eLife.47395.042

could promote cell survival in the presence of noxious stimuli. To test this hypothesis, we exposed neutrophil-like PLB-985 cells to the highly immunogenic mycobacterial protein antigen Ag85B and staurosporine in the presence of native or HOCl-treated HSA. The mycolyltransferase Ag85B is the major antigen produced and secreted from all mycobacterial species during infection (*Belisle et al., 1997*) and has been shown to play an important role in the induction of protective immunity (*Andersen, 1994*; *Baldwin et al., 1998*) by inducing strong T cell proliferation and IFN-γ secretion (*Launois et al., 1994*; *Takamura et al., 2005*). Activation of neutrophils during mycobacterial infections is often accompanied by accelerated apoptosis (*Alemán et al., 2002*; *Perskvist et al., 2002*), but the mechanism by which mycobacterial species or their secreted antigens induce apoptosis has not been elucidated in detail. Staurosporine, a protein kinase inhibitor, has been characterized as an efficient inducer of apoptosis in various cell types via caspase-dependent and -independent pathways (*Chae et al., 2000*; *Zhang et al., 2004*) and thus, was also used as an cell death-promoting agent in our experiments.

After 1 hr or 6 hr of incubation with Ag85B and staurosporine, respectively, viability of the variously treated cells was evaluated by flow cytometry using Annexin V-FITC/propidium iodide (PI) staining. During early apoptosis, phosphatidylserine is translocated from the inner to the outer cell membrane leaflet with the plasma membrane left intact and thus, available for binding of extrinsically applied annexin V protein (*Vermes et al., 1995*). In late apoptosis/necrosis, the integrity of the plasma membrane is lost, allowing the normally membrane-impermeable propidium iodide to enter and stain the DNA. Cells that are in late apoptosis or necrotic are thus both Annexin V-FITC and PI positive. Accordingly, cells that are viable are both Annexin V-FITC and PI negative.

Data plots were generated from analysis of ungated data (*Figure 8*). Viable cells appear in the lower left quadrant (Q4), early apoptotic cells in lower right quadrant (Q3) and late apoptotic/necrotic cells in the upper right (Q2) quadrant.

Upon treatment with Ag85B, the ratio of viable cells to early/late apoptotic cells has markedly decreased compared to the control cells, that were incubated in the absence of Ag85B, pointing toward a lifetime-limiting effect of Ag85B. Pretreatment with the broad-spectrum pan-caspase inhibitor Z-VAD-FMK did not significantly affect Ag85B-induced cell death, suggesting that Ag85B exerts its lethal effect through a caspase-independent mechanism. To test whether the toxicity of Ag85B depends on its uptake by the immune cells, we pretreated the cells with cytochalasin D, an inhibitor of phagocytosis, prior to the addition of Ag85B. Reduction of the cells' phagocytic capacity resulted in markedly improved cell survival in the presence of Ag85B. This phenomenon is illustrated by a shift of the cells from Q3 to Q4 on the Annexin V-FITC/PI plot with concomitant reduction of necrotic cells. Remarkably, pretreatment with HOCl-modified HSA, added at a 155-fold molar excess over Ag85B, completely prevented cell death upon addition of Ag85B. In contrast, native HSA had

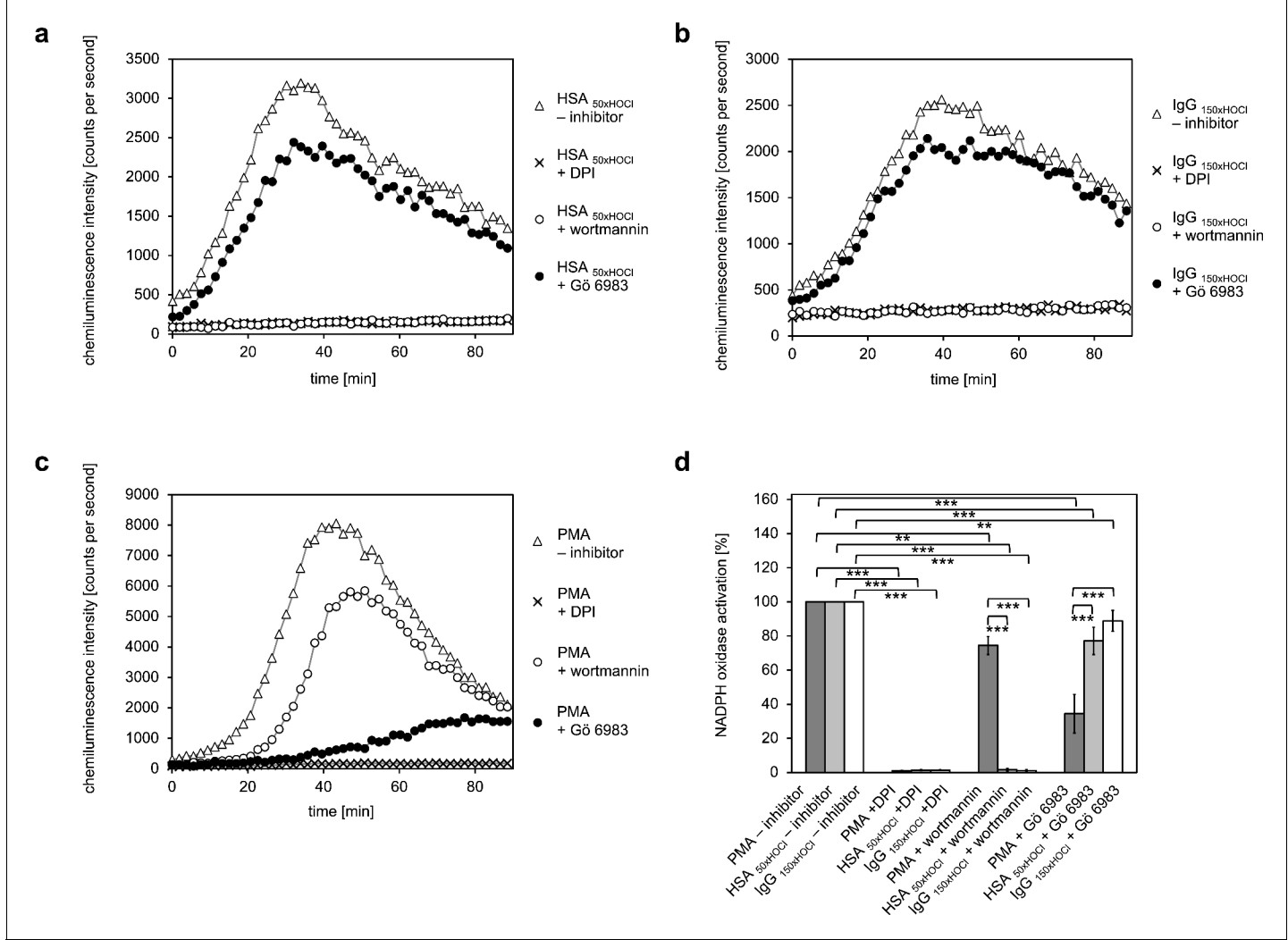

**Figure 7.** Activation of the NADPH oxidase of neutrophil-like cells by HOCl-treated serum albumin and immunoglobulin G occurs predominantly via a PI3K-dependent signaling pathway. Effect of 10 μM diphenyleneiodonium (DPI; NADPH oxidase inhibitor), 100 nM wortmannin (PI3K inhibitor) and 200 nM Gö 6983 (protein kinase C (PKC) inhibitor) on the NADPH oxidase activation mediated by (a) 3 mg · mL$^{-1}$ HSA $_{50xHOCl}$, (b) 3 mg · mL$^{-1}$ IgG $_{150xHOCl}$ and (c) 0.2 μM PMA was tested. (d) Results shown in a, b and c are expressed as integrated total counts (means and standard deviations of three independent measurements) higher than the respective buffer control. Student's t-test: **p<0.01, ***p<0.001.

DOI: https://doi.org/10.7554/eLife.47395.043

The following source data and figure supplements are available for figure 7:

**Source data 1.** Numerical chemiluminescence plate reader data represented in *Figure 7a and d*.
DOI: https://doi.org/10.7554/eLife.47395.046

**Source data 2.** Numerical chemiluminescence plate reader data represented in *Figure 7b and d*.
DOI: https://doi.org/10.7554/eLife.47395.047

**Source data 3.** Numerical chemiluminescence plate reader data represented in *Figure 7c and d*.
DOI: https://doi.org/10.7554/eLife.47395.048

**Figure supplement 1.** Addition of catalase, superoxide dismutase (SOD) or both inhibited immune cell-induced chemiluminescence of lucigenin.
DOI: https://doi.org/10.7554/eLife.47395.044

**Figure supplement 1—source data 1.** Numerical chemiluminescence plate reader data represented in *Figure 7—figure supplement 1*.
DOI: https://doi.org/10.7554/eLife.47395.045

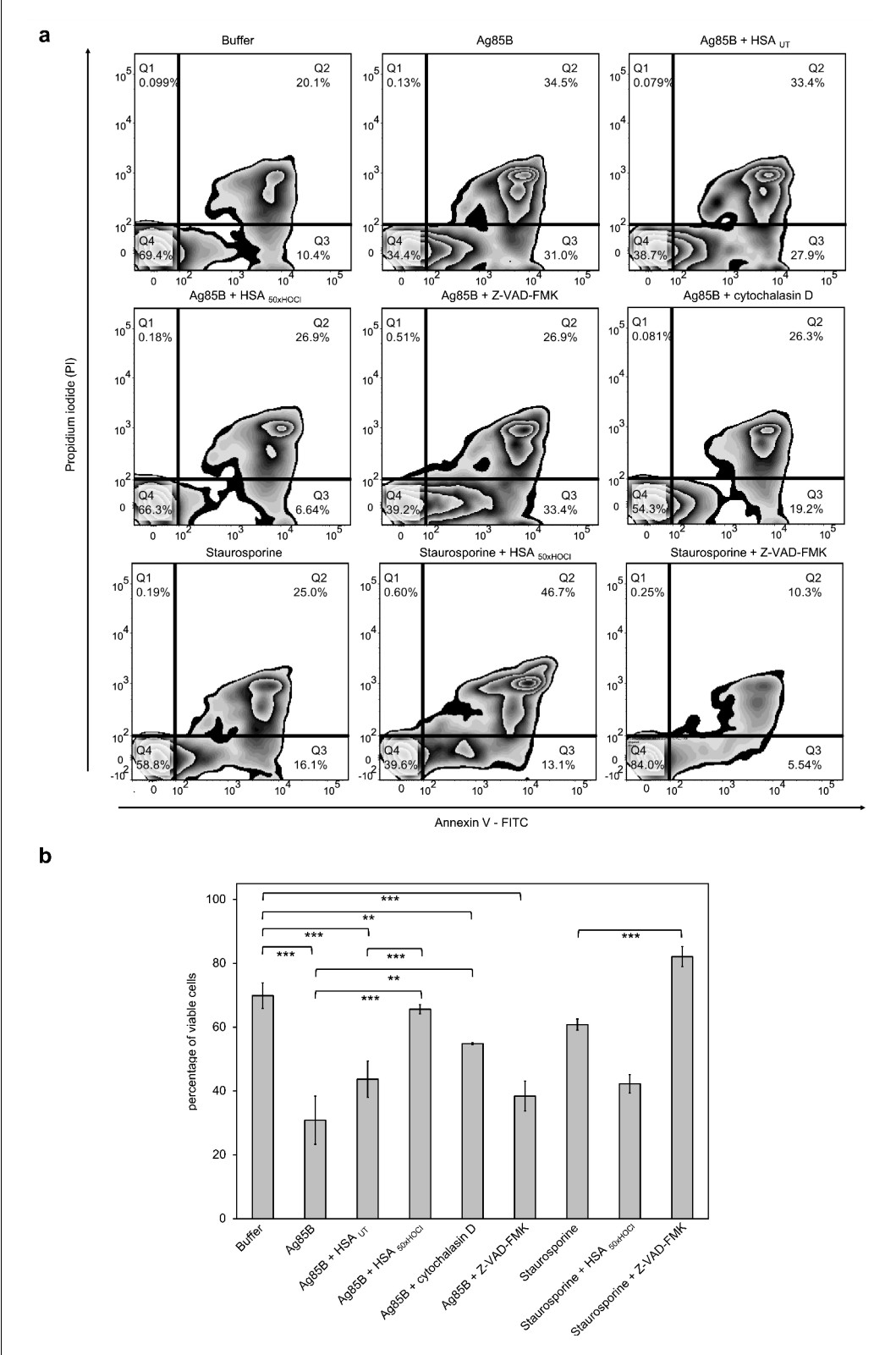

**Figure 8.** HOCl-treated serum albumin improves survival of neutrophil-like cells in the presence of the major mycobacterial protein antigen Ag85B. Differentiated neutrophil-like PLB-985 cells were preincubated with 50 µM Z-VAD-FMK, 155 µM native (HSA $_{UT}$) or HOCl-treated HSA (HSA $_{50xHOCl}$) prior to the addition of 1 µM Ag85B or 2 µM staurosporine. After 1 hr (Ag85B) or 6 hr (staurosporine) of incubation, viability of the variously treated cells was assessed by flow cytometry using Annexin V/propidium iodide (PI) staining. Cells treated only with buffer served as control. (a) Annexin V-FITC vs.

*Figure 8 continued on next page*

*Figure 8 continued*

propidium iodide dot plots show all analyzed events. Staining of cells simultaneously with Annexin V- FITC (green fluorescence) and the non-vital dye PI (red fluorescence) allows the discrimination of viable, intact cells (FITC-, PI-; Q4), early apoptotic (FITC+, PI-; Q3) and late apoptotic/necrotic cells (FITC +, PI+; Q2). 20,000 events were acquired and recorded per sample. Data were analyzed using FlowJo (version 10) software. Results shown are representative of three experiments. (b) Results of three independent experiments for viable cells are shown (means and standard deviations). Student's t-test: *p<0.05, **p<0.01, ***p<0.001.

DOI: https://doi.org/10.7554/eLife.47395.049

The following source data is available for figure 8:

**Source data 1.** Numerical flow cytometry data represented in *Figure 8a and b*.

DOI: https://doi.org/10.7554/eLife.47395.050

almost no effect on Ag85B-induced cell death. We thus speculated that HOCl-treated HSA could rescue immune cells from Ag85B-induced cell death by preventing or strongly reducing its uptake, rather than by boosting PI3K/Akt signaling. In support of this conclusion, Z-VAD-FMK, but not HOCl-treated HSA was able to significantly reduce staurosporine-induced apoptosis. Instead, when combined with staurosporine, HOCl-modified HSA triggered necrosis with the majority of the cells being Annexin V and PI positive.

## HOCl-treated serum albumin effectively binds Ag85B and reduces its uptake by neutrophil-like cells

Prompted by the finding that HOCl-treated HSA can act as a chaperone being highly effective at preventing protein aggregation, we asked whether N-chlorinated HSA can also bind to Ag85B and thus prevent its uptake by immune cells. To test this, Ag85B was diluted stepwise in the presence of native or HOCl-treated HSA and aggregation of Ag85B was monitored by light scattering. When denatured Ag85B was diluted into buffer, it readily formed aggregates (*Figure 9a,b*). This aggregation could not be prevented by untreated HSA. Presence of HOCl-modified HSA, however, significantly reduced the aggregation of Ag85B when added at a 80-fold molar excess over the protein. Again, reduction with ascorbate fully abrogated HSA $_{HOCl}$'s ability to bind Ag85B.

We argued that Ag85B's association with HOCl-modified HSA could decrease its propensity to enter immune cells and thus, provide a possible explanation for the enhanced survival of neutrophils in the presence of this mycobacterial antigen. To test whether this HSA $_{HOCl}$-mediated cell survival is indeed due to a reduced uptake, we added recombinant fluorescently labeled Ag85B (Ag85B-488) to differentiated neutrophil-like cells and analyzed its uptake in the presence of native or HOCl-treated HSA using flow cytometry. Cells co-incubated with native HSA accumulated around 30% less Ag85B compared to cells treated with Ag85B alone (*Figure 9c,d*). But presence of HOCl-treated HSA reduced uptake of Ag85B by up to 80%. Sustained cell viability in the presence of Ag85B is thus most likely linked to decreased antigen uptake by the immune cells.

## Discussion

In our study, we found that a number of proteins in plasma turned into chaperone-like proteins when treated with HOCl. These proteins are highly effective at preventing formation of potentially toxic protein aggregates in vitro. The role of plasma proteins in inflamed tissue thus goes beyond that of a passive sink for HOCl.

Since exposure to high HOCl concentrations can lead to oxidative, irreversible protein unfolding (*Heinecke et al., 1993*), one might speculate that the observed chaperone activity of such unfolded plasma proteins could simply be the result of an increased affinity to other unfolded proteins. Here, we provide direct evidence that HOCl-mediated conversion of plasma proteins into potent chaperones depends primarily on reversible chlorination of their basic amino acids. Along this line, treatment of HSA with HOCl led to dose-dependent reduction of the accessible amino group content accompanied by an increase in N-chloramine content and surface hydrophobicity, providing an obvious explanation for the increased affinity to unfolding proteins of N-chlorinated plasma proteins. Since N-chlorination does not irreversibly alter the structural and functional properties of a protein, this reversible post-translational modification provides a new strategy to recruit new chaperone-like

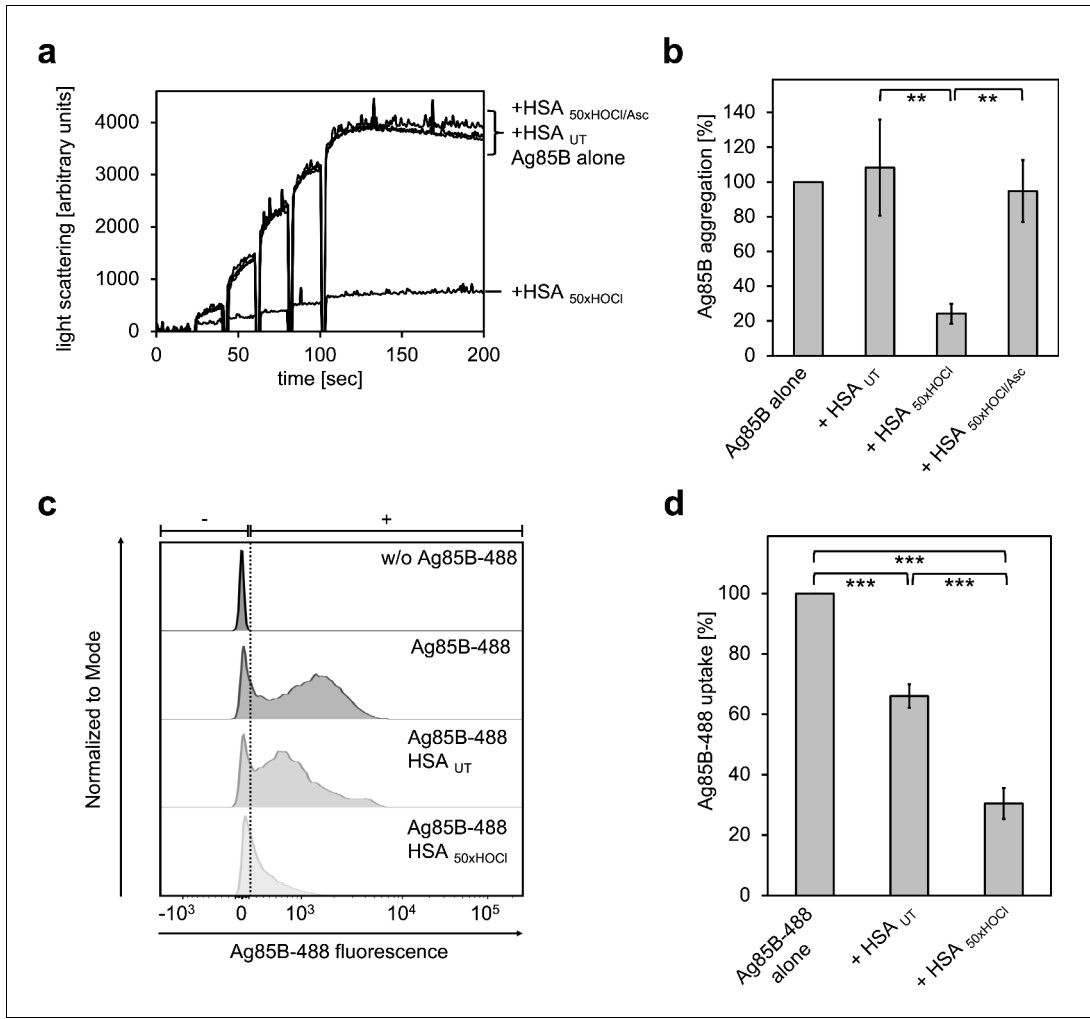

**Figure 9.** HOCl-treated serum albumin binds to and prevents uptake of the major mycobacterial protein antigen Ag85B by neutrophil-like cells. (**a, b**) HSA, treated with a 50-fold molar excess of HOCl (HSA $_{50xHOCl}$) significantly decreased aggregation of denatured Ag85B as measured by light scattering at 360 nm. Reduction of HSA $_{50xHOCl}$ with a 50-fold molar excess of the antioxidant ascorbate (HSA $_{50xHOCl/Asc}$) reversed this chaperone activity. (**a**) A representative measurement of Ag85B aggregation in the presence of native HSA (HSA $_{UT}$), HSA $_{50xHOCl}$ and HSA $_{50xHOCl/Asc}$ is shown. Labels of aggregation curves are written in the order of the final intensity of light scattering of the respective treatment. (**b**) Data are represented as means and standard deviations from three independent aggregation assays. Aggregation of Ag85B in the absence of HSA was set to 100% and all the data are presented as percentage of this control. (**c, d**) Differentiated neutrophil-like PLB-985 cells were incubated in the absence or presence of the fluorescently-labeled Ag85B protein, Ag85B-488. In some cases, HSA $_{UT}$ or HSA $_{50xHOCl}$ was added at a 50-fold molar excess over Ag85B-488 to the cells. After one hour of incubation, uptake of fluorescent Ag85B-488 by the variously treated cells was assessed by flow cytometry. 30,000 events were acquired and recorded per sample. Data were analyzed using FlowJo (version 10) software. (**c**) Single-parameter histogram overlays of Ag85B-488 fluorescence of the various samples are shown. (**d**) Results of three independent experiments are shown (means and standard deviations). Student's t-test: **p<0.01, ***p<0.001. Uptake of Ag85B-488, reflected by the median fluorescence intensity, in the absence of HSA was set to 100%.

DOI: https://doi.org/10.7554/eLife.47395.051

The following source data is available for figure 9:

**Source data 1.** Numerical light scattering data obtained during protein aggregation assays represented in *Figure 9a and b*.

DOI: https://doi.org/10.7554/eLife.47395.052

**Source data 2.** Numerical flow cytometry data obtained during protein aggregation assays represented in *Figure 9c and d*.

DOI: https://doi.org/10.7554/eLife.47395.053

proteins on demand in response to HOCl-stress in order to minimize self-damage associated with the formation of protein aggregates during inflammation.

In recent years, other roles for HOCl-modified plasma proteins and lipoproteins in inflammatory processes have also been described. For example, HOCl-oxidized low-density lipoproteins (*Kopprasch et al., 1998*), as well as HOCl-modified HSA (*Gorudko, 2014*; *Witko-Sarsat et al., 1998*) have been shown to elicit various polymorphonuclear leukocyte (PMNL) responses such as the NADPH-dependent generation of reactive oxygen species (ROS), degranulation or shape change (*Gorudko, 2014*), but the mechanism for the HOCl-mediated functional conversion into a potent activator of human PMNLs remained unclear.

As *N*-chlorination was the principal chemical modification responsible for the plasma protein's switch to a chaperone-like holdase, we hypothesized that it could also be the reason for the activation of immune cells. To mimic an in vivo situation, where plasma proteins are located in immediate vicinity to accumulated neutrophils in inflamed tissue, we exposed several plasma fractions to HOCl at pathophysiological concentrations (*Summers et al., 2008*) and investigated the effect of the resulting products on ROS production by neutrophil-like cells. Remarkably, we found that not only the main plasma protein HSA, but the majority of plasma fractions tested, when treated with a sufficient amount of HOCl, were able to elicit a significant immune response, as shown by the increased generation of ROS by the immune cells. This effect was not seen upon re-reduction of the HOCl-modified proteins or prior methylation of their basic amino acids, strongly supporting an *N*-chlorination based mechanism as well. To the best of our knowledge, this is the first study which demonstrates that reversible HOCl-mediated *N*-chlorination is the principal mechanism of turning plasma proteins into critical modulators of the innate immune response.

*Gorudko (2014)* reported that the phosphoinositide 3-kinase (PI3K) inhibitor wortmannin inhibited the stimulating effect of HOCl-modified HSA on immune cells. The same was true in our model system: the immunomodulatory action of both HOCl-modified HSA and immunoglobulin G could be fully inhibited or strongly attenuated by wortmannin. PI3K and its downstream effectors, such as the serine/threonine kinase Akt, are indeed considered to play a key role in the regulation of the neutrophil NADPH oxidase (*Kennedy and DeLeo, 2008*; *Hawkins et al., 2007*). Since the PKC inhibitor Gö 6983 showed only little effect on HSA $_{50xHOCl}$- and IgG $_{150xHOCl}$-mediated NADPH oxidase activation, we thus propose that these proteins stimulate the neutrophil respiratory burst predominantly via PI3K-dependent signaling pathways. However, further studies are needed to investigate the exact mechanism by which these proteins trigger PI3K signaling. A possible scenario might be, that *N*-chlorination confers higher affinity to a membrane scavenger receptor allowing the binding of HOCl-modified HSA. In support of this theory, it has been reported that *N*-chlorinated, but not native HSA can irreversibly bind to and block the major high-density lipoprotein receptor, scavenger receptor class B, type 1 (SR-BI) (*Binder et al., 2013*). It is therefore tempting to speculate that *N*-chlorination could also increase the affinity of IgG to Fc gamma receptors (FcγRs).

The PI3K/Akt pathway is also considered an anti-apoptotic pathway (*Zhuang et al., 2011*; *Chang et al., 2003*). Thus, it was tempting to hypothesize that HOCl-modified HSA may play a role as a pro-survival molecule during inflammation. Remarkably, we found that HOCl-modified HSA indeed enhances survival of neutrophils in the presence of the highly immunogenic mycobacterial protein antigen Ag85B, however, not in the presence of staurosporine, a broad-spectrum protein kinase inhibitor that induces apoptosis in various cell types (*Chae et al., 2000*; *Zhang et al., 2004*). Looking for a mechanism by which HOCl-modified HSA is able to rescue immune cells from Ag85B-induced cell death, we found that it can effectively bind to and strongly decrease the phagocytosis of Ag85B. Internalization of Ag85B by the cells proved to be the direct cause of cell death and could also be prevented by other inhibitors of phagocytosis.

Neutrophils are typically short-lived, but their apoptosis can be delayed both by microbial products and by various proinflammatory stimuli (*Lee et al., 1993*; *Colotta et al., 1992*). In this study, we describe HOCl-modified HSA as a novel pro-inflammatory mediator, which can promote cell survival by binding to highly immunogenic foreign antigens and reducing their phagocytosis at sites of bacterial infection. A similar phenomenon was observed in other studies, where HOCl-modified HSA was shown to bind and neutralize proteins from HIV and West Nile virus (*Vossmann et al., 2008*; *Speth et al., 2013*).

The immunomodulatory effects of *N*-chlorinated plasma proteins found in this and previous studies constitute a double-edged sword. Although stimulation of neutrophil respiratory burst and

enhanced neutrophil survival may be beneficial for pathogen elimination at the initial stage of infection, it can eventually perpetuate a positive feedback loop and contribute to the development and progression of chronic inflammation (*Figure 10*). Secretion of HOCl and other oxidants by permanently activated neutrophils leads not only to the destruction of neighboring, healthy cells resulting in tissue injury (*Pechous, 2017*; *Weiss, 1989*; *Margaroli et al., 2019*), but generates more *N*-chlorinated plasma proteins. Similarly, the pro-survival effect of *N*-chlorinated HSA is not only positive, as cell death and the subsequent recognition of dying neutrophils by macrophages has a critical function in the resolution of the inflammatory response and is strictly required to protect the surrounding tissue and prevent pathological sequelae (*Erwig and Henson, 2007*). Were all these properties of AOPPs dependent on the numerous irreversible modifications reportedly caused by HOCl exposure,

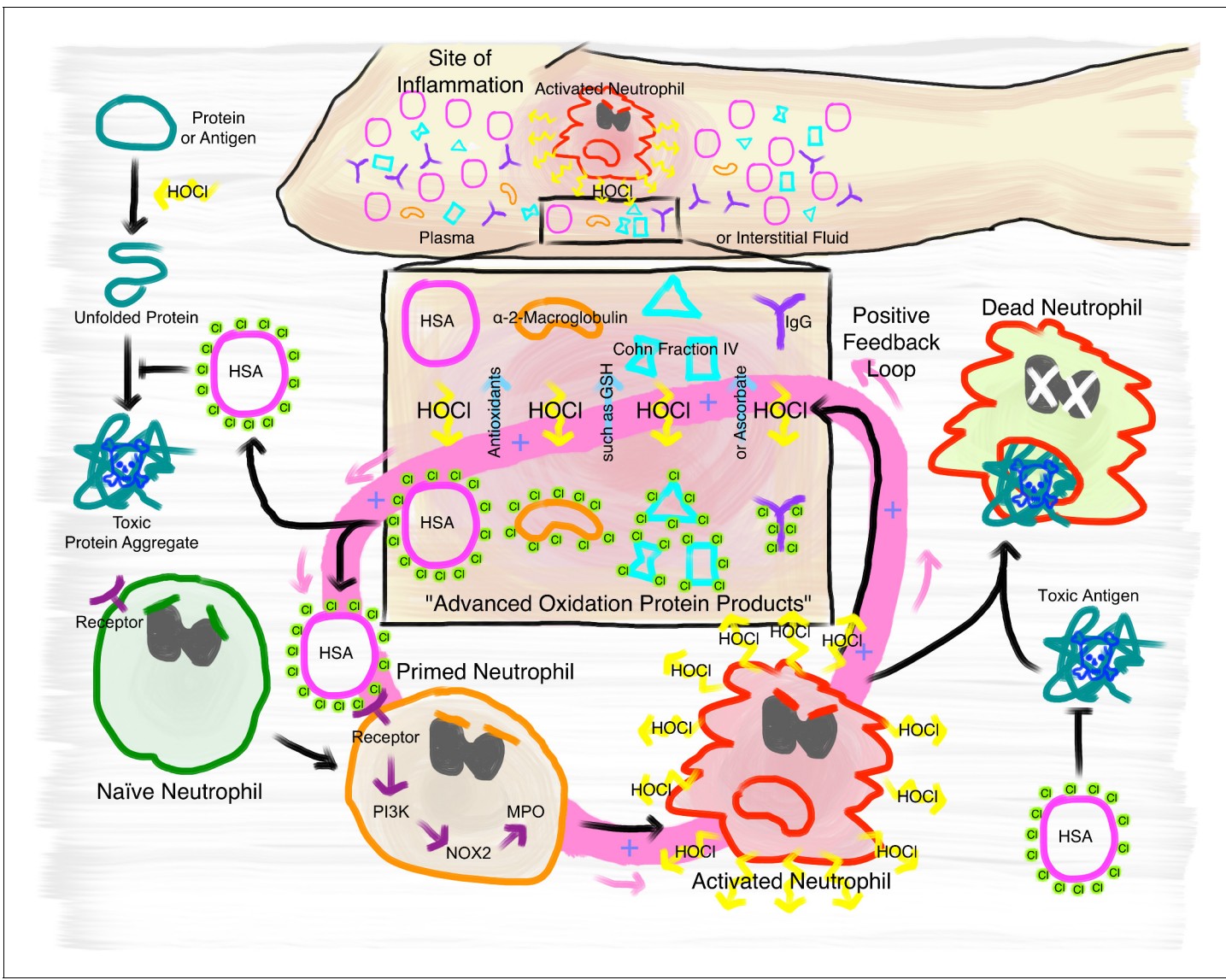

**Figure 10.** Proposed mechanism of the immunomodulatory role of *N*-chlorination of plasma proteins at a site of inflammation. At the site of inflammation neutrophils (and potentially other immune cells) are activated. Neutrophils then produce HOCl at concentrations of up to 25 to 50 mM per hour. Plasma proteins, such as HSA, α- and β-globulins (Cohn Fraction IV), γ-globulins (IgG), and α2-Macroglobulin then act as an effective sink. Reversible *N*-chlorination of these proteins turns them into effective chaperons, which can prevent the formation of protein aggregates and their uptake by immune cells, enhancing the survival of neutrophils in the presence of toxic antigens. *N*-chlorinated plasma proteins also activate more immune cells, which then in turn produce more HOCl, leading to the formation of more *N*-chlorinated plasma proteins. This positive feedback loop can be attenuated and deactivated by antioxidants present in plasma, such as ascorbate and reduced glutathione (GSH).
DOI: https://doi.org/10.7554/eLife.47395.054

this could lead to a spreading out-of-control immune reaction. The presence of antioxidants such as ascorbate and glutathione in plasma, both of which can reduce *N*-chlorination, provide a mechanism to contain the immune reaction at the site of inflammation. While these antioxidants are probably quickly depleted in the direct vicinity of an acute inflammation, they could inactivate *N*-chlorinated plasma proteins that escape the site of inflammation, if present in sufficient quantity. Indeed, depletion of these antioxidants is often associated with chronic inflammation and other diseases (*Bonham et al., 1999*; *Ruan et al., 1997*; *Huijskens et al., 2016*; *Rose et al., 2012*) and antioxidant therapy such as high-dose intravenous vitamin C treatment leads to decrease of inflammation markers such as CRP (*Mikirova et al., 2012*; *Carr and Cook, 2018*; *Block et al., 2009*). Our data explains important aspects of these effects and further highlights the role of antioxidant homeostasis in inflammatory processes.

In summary, our data support a critical role for HOCl-mediated *N*-chlorination of plasma proteins during inflammatory processes and suggest that it is the critical modification mediating the physiological effects of so-called AOPPs. Although the reversible conversion of HOCl-modified plasma proteins to effective chaperones confers protection against HOCl-induced protein aggregation, the increase of their immunogenicity can potentially exacerbate self-damage at sites of inflammation through a positive feedback loop. The fact that the activation of immune cells is mediated through *N*-chlorination, a modification that is reversible by antioxidants present in plasma, provides a mechanism to attenuate or deactivate this positive feedback loop. These findings contribute importantly to our understanding of the development and progression of chronic inflammation.

# Materials and methods

## Preparation of plasma protein solutions

Albumin from human serum (HSA, Product # A9511), human γ-globulins (Product # G4386), immunoglobulin G from human serum (IgG, Product # I4506) Cohn fraction IV (Product # G3637) and human whole serum (sterile-filtered from male AB plasma, Product # H4522) were purchased from Sigma-Aldrich, St. Louis, USA, and used without further purification.

Protein stock solutions were freshly prepared by dissolving or diluting varying amounts of the protein fractions in 1xPBS buffer, pH 7.4 (Gibco Life Sciences).

## Purification of α₂-macroglobulin

α₂-Macroglobulin was purified from human plasma (obtained from Zen-Bio, Inc, North Carolina, USA, Product # SER-SLP, Lot # 11108) by the method of *Imber and Pizzo (1981)* with slight modifications. Briefly, 880 mL fresh-frozen human plasma were thawed on ice and dialyzed against frequent changes of deionized water for 72 hr at 4°C using a Spectra/Por dialysis membrane with a MWCO of 12–14 kDa (Spectrum Laboratories Inc, Rancho Dominguez, CA). Insoluble material was removed by centrifugation for 30 min at 10,000 x g and 4°C. The supernatant plasma solution (200 mL) was then dialyzed for 24 hr at 4°C against 5 L 1xPBS pH 6.0. Metal chelate chromatography was performed at 4°C using an IMAC zinc-Sepharose 6 Fast Flow column (2.6 × 20 cm, GE Healthcare Life Sciences, Amersham, UK) equilibrated with 1xPBS pH 6.0. Dialyzed plasma was applied to the column and washed with 1xPBS pH 6.0 until the absorption at 280 nm of the eluant (measured with a JASCO V-650 UV/VIS spectrophotometer (JASCO, Tokyo, Japan)) reached a value lower than 0.01. Bound protein was then eluted from the column with 0.01 M NaOAc, 0.15 M NaCl, pH 5.0. Peak protein fractions were combined and concentrated using Vivaspin 20 concentrators with a MWCO of 100 kDa. Gel filtration of the concentrated protein pool fraction was performed at 4°C on a HiPrep 26/60 Sephacryl S-300 High-resolution column (2.6 × 60 cm) equilibrated with 1xPBS pH 7.4. High-molecular-weight peak fractions containing α₂-macroglobulin were combined, concentrated and dialyzed against 1xPBS pH 7.4 containing 40% (v/v) glycerol. Aliquots were stored at −20°C.

## Determination of protein concentrations

Protein concentration in g. L$^{-1}$ of human serum was calculated using Pierce Bicinchoninic Acid (BCA) Protein Assay Kit (Thermo Fisher Scientific, Waltham, MA) with bovine albumin as standard carried out following the manufacturer's instructions. To calculate a molar concentration, an average molar mass of proteins of 66,357.12 Da was assumed (molar mass of human serum albumin).

For HSA, IgG, the γ-globulin fraction, and $\alpha_2$-macroglobulin, the concentration was determined by measuring the absorbance at 280 nm ($A_{280nm}$) using a JASCO V-650 UV/VIS spectrophotometer. The molar extinction coefficient used for HSA was $\varepsilon_{280}$ = 35,700 $M^{-1}$ $cm^{-1}$ [98]. Concentration of the γ-globulin fraction was estimated using the extinction coefficient for immunoglobulin G (IgG) of 1.36 $cm^{-1}$ $(mg. mL^{-1})^{-1}$ [99]. Assuming a molecular weight of 150,000 Da (*Murphy and Weaver, 2016*), the molar extinction coefficient at 280 nm used for was 210,000 $M^{-1}$ $cm^{-1}$. The molar extinction coefficient used for $\alpha_2$-macroglobulin, $\varepsilon_{280}$ = 145,440 $M^{-1}$ $cm^{-1}$, was calculated from amino acid sequence using ProtParam (*Wilkins et al., 1999*).

Concentration of Cohn fraction IV was determined using Pierce Bicinchoninic Acid (BCA) Protein Assay Kit and bovine serum albumin (BSA) as standard according to the manufacturer's instructions. To calculate a molar concentration, a weighted average protein mass of 80,000 Da was assumed based on the composition of the Cohn fraction IV.

## Methylation of proteins

Proteins were methylated as previously described (*Rypniewski et al., 1993*) with some modifications. Proteins were dissolved in 1 ml 1xPBS pH 7.4 to a concentration of 10 mg. $mL^{-1}$ and the solution cooled to 4°C. 20 µl of 60 mg $mL^{-1}$ dimethylamine borane complex and 40 µl 1M formaldehyde were then added. After 2 hr of incubation at 4°C, this step was repeated. 2 hr later a final aliquot of 10 µl dimethylamine borane complex solution was added, before incubation of the reaction mixture at 4°C overnight. The next morning 125 µl of 1M Tris pH 7.5 were added to stop the reaction. The reacting agents were then separated from the now methylated proteins by size-exclusion chromatography using Nap-5 columns according to the manufacturer's instructions (GE Healthcare Life Sciences, Amersham, UK).

## Treatment of proteins with HOCl and $H_2O_2$

The concentration of the NaOCl stock solution of 0.64 M (Sigma-Aldrich, St. Louis, MO) was confirmed regularly by measuring the absorbance at 292 nm using a JASCO V-650 UV/VIS spectrophotometer and the extinction coefficient $\varepsilon_{292}$ = 350 $M^{-1}$ $cm^{-1}$. When necessary, NaOCl stock solution was diluted by mixing an adequate volume of NaOCl with 1x PBS solution pH 7.4 immediately prior to each chlorination reaction. A 30% (w/w) stock solution of $H_2O_2$ in $H_2O$ (Sigma-Aldrich, St. Louis, MO) was diluted in 1x PBS solution pH 7.4 to a final concentration of 0.64 M immediately prior to use.

Varying amounts of a protein were then treated with various molar excesses of NaOCl or $H_2O_2$, ranging from 10-fold to 150-fold, for 10 min at 30°C (maximum NaOCl concentration used was 50 mM – a concentration that can be produced by neutrophils per hour in chronically inflamed tissues; *Summers et al., 2008*).

Excess HOCl or $H_2O_2$ was removed by size-exclusion chromatography using Nap-5 columns according to the manufacturer's instructions. Due to dilution during the Nap-5 desalting step, protein concentrations were re-determined as described above.

## Reduction of HOCl-treated proteins

To reverse protein *N*-chlorination in HOCl-treated proteins, sodium ascorbate, dithiothreitol (DTT) or methionine was dissolved in 1xPBS pH 7.4 to a concentration of 1 M (sodium ascorbate, DTT) or 140 mM (methionine) and the proteins were incubated with a 50-fold molar excess of these reductants for 45 min at 37°C. After removal of excess reductant (see above), protein concentrations were again re-determined.

## Preparation of taurine chloramines

Taurine monochloramine and taurine dichloramine were prepared fresh daily by a dropwise addition of 50 µM and 10 mM NaOCl, respectively, to 5 mM taurine (Sigma-Aldrich, St. Louis, USA) in 1x PBS solution pH 7.4 with continuous stirring for 10 min at 30°C. Presence of taurine monochloramine ($\lambda_{max}$ is 252 nm), dichloramine ($\lambda_{max}$ is 300 nm) and unreacted NaOCl ($\lambda_{max}$ is 292 nm) was monitored by UV absorption spectra using a JASCO V-650 UV/VIS spectrophotometer (*Marcinkiewicz et al., 2006*; *Thomas et al., 1986*). Taurine monochloramine concentration was determined using the molar extinction coefficient $\varepsilon_{252}$ = 415 $M^{-1}$ $cm^{-1}$(*Thomas et al., 1986*).

## Purification of *E. coli* IlvA

*E. coli* IlvA was overexpressed and purified as previously described (*Müller et al., 2014*). Protein concentration was determined using a JASCO V-650 UV/VIS spectrophotometer and the extinction coefficient $\varepsilon_{280}$ = 31,860 M$^{-1}$ cm$^{-1}$.

## Protein aggregation assays with citrate synthase and IlvA

200 µL Citrate synthase in ammonium sulfate solution (Sigma-Aldrich, St. Louis) was dialyzed overnight against 2 L 20 mM Tris 2 mM EDTA buffer at 4°C under constant stirring using a Spectra/Por dialysis membrane with a MWCO of 6–8000 Da. (Spectrum Laboratories Inc, Rancho Dominguez, CA). This dialyzed citrate synthase preparation or purified IlvA was then chemically denatured in 4.5 M GdnHCl, 40 mM HEPES, pH 7.5 at room temperature overnight. The final concentration of denatured citrate synthase or IlvA was 12 µM.

Aggregation was induced by the addition of 20 µL denatured citrate synthase or IlvA stock to 1580 µL 40 mM HEPES/KOH buffer, pH 7.5. Final concentration of citrate synthase or IlvA in the aggregation assay was thus 0.15 µM. Untreated or treated plasma proteins were added to the assay buffer to a final concentration of 0.5 µM ($\alpha_2$-Macroglobulin) or 1.5 µM (HSA, IgG, γ-globulins, Cohn fraction IV) (corresponding to a 3.3-fold and 10-fold molar excess over the dimeric citrate synthase, respectively) prior to the addition of citrate synthase. For the IlvA aggregation assay, untreated HSA or HSA treated with a 50-fold molar excess of HOCl was added to the assay buffer to a final concentration of 1.5 µM, corresponding to a 10-fold molar excess over IlvA. Various taurine chloramine preparations (see above) were added to a final concentration of 1.5 µM or 148.5 µM, corresponding to a 10-fold and 990-fold molar excess over the dimeric citrate synthase, respectively. The increase of light scattering was monitored in a JASCO FP-8500 fluorescence spectrometer equipped with an EHC-813 temperature-controlled sample holder (JASCO, Tokyo, Japan) at 30°C for 200–240 s under continuous stirring. Measurement parameters were set to 360 nm (Ex/Em), 2.5 nm slit width (Ex/Em) and medium sensitivity. Chaperone activity was expressed as the difference between initial and final light scattering of an individual sample in arbitrary units. Aggregation of citrate synthase or IlvA in the absence of any other proteins was set to 100%. Depending on the batch of citrate synthase, absolute maximum and minimum light scattering values may vary, thus control experiments with the same batch were carried out for each individual aggregation experiment.

## Detection of accessible amino groups in proteins

Accessible amino groups in HSA were detected using fluorescamine (Sigma-Aldrich, St. Louis) as described (*Udenfriend et al., 1972*). Briefly, 334 µl of 3 mg. mL$^{-1}$ fluorescamine stock in acetone were added to 1 mL of 80 µg. mL$^{-1}$ native HSA or the variously treated HSA solutions described above. Emission spectrum of fluorescamine from 400 to 600 nm was measured upon excitation with 388 nm using a JASCO FP-8500 fluorescence spectrometer. The relative amount of accessible amino groups in the variously treated HSA samples was calculated by setting the maximum fluorescence of native HSA to 100% representing the total relative amino group content.

## Quantification of chloramines using 3,3',5,5'-tetramethylbenzidine

Quantification of chloramines on variously treated HSA samples was performed as described by *Dypbukt et al. (2005)* with slight modifications. Briefly, 250 µM HOCl were added under rigorous stirring at room temperature to 25 mM taurine in 1x PBS buffer solution, pH 7.4. After 10 min of incubation, taurine monochloramine concentration was determined as described above. For the construction of a monochloramine standard curve, taurine monochloramine was diluted in 1x PBS to a final concentration of 10, 20, 30 and 40 µM. Subsequently, 1 mL of these dilutions was rapidly mixed with 250 µL developing reagent composed of 2 mM 3,3',5,5'-tetramethylbenzidine (TMB; Sigma-Aldrich, St. Louis) and 100 µM sodium iodide (Sigma-Aldrich) in 400 mM acetate buffer, pH 5.4, containing 10% (v/v) dimethylformamide (Sigma-Aldrich). At the same time, 250 µL developing reagent were added to 1 mL of variously treated HSA preparations. Untreated HSA and HOCl-treated HSA after reduction with sodium ascorbate, DTT or methionine were used at a final concentration of 10 µM. HSA treated with a 10-fold or 50-fold molar excess of HOCl was used at a final concentration of 2.5 µM and 5 µM or 0.5 µM and 1 µM, respectively. After 30 min of incubation, absorbances at 652 nm were measured with a JASCO V-650 spectrophotometer. For each taurine monochloramine

concentration the average absorbance value was calculated. These values were then plotted against the respective monochloramine concentration and a best-fit straight line was drawn through these points using the trendline function in Microsoft Excel Version 16.20. The line slope equation derived from linear regression was used to calculate the amount of monochloramines on the variously treated HSA samples. Absolute number of monochloramines per molecule HSA was calculated by dividing the measured monochloramine concentration by the respective protein concentration.

## Nile red hydrophobicity assay

Nile red (Sigma-Aldrich, St. Louis, USA) was dissolved in dimethyl sulfoxide (DMSO) to a final concentration of 30 µM. Varying amounts of native or HOCl-treated HSA (0–200 µM) in 1xPBS were mixed with Nile red stock to a final dye concentration of 1.6 µM. Fluorescence was measured using a JASCO FP-8500 fluorescence spectrometer with the following parameters: 550 nm excitation, 570–700 nm emission, 5 nm slit width (Ex/Em) and medium sensitivity. Concentrations of native or HOCl-treated HSA, at which the proteins were half-saturated with dye were calculated by plotting the fluorescence intensity of Nile red against the logarithm of the molar protein concentrations. Data were fitted with GraphPad Prism eight software using the sigmoid fit function.

## PLB-985 culture and differentiation

The human myeloid leukemia cell line PLB-985 (certified mycoplasma negative, obtained from DSMZ, German collection of microorganisms and cell culture) was cultured in RPMI-1640 medium supplemented with 10% heat-inactivated FBS and 1% GlutaMAX (Life Technologies, Carlsbad, CA) at 37°C with 5% $CO_2$. Cells were passaged twice weekly to maintain a cell density between $2 \times 10^5$ and $1 \times 10^6 \cdot mL^{-1}$ and used until passage no. 10. For differentiation into neutrophil-like cells, cells were seeded at a density of $2 \times 10^5 \cdot mL^{-1}$ and cultured for 96 hr in RPMI-1640 medium with 10% FBS, 1% GlutaMAX and 1.25% DMSO (*Pivot-Pajot et al., 2010*). After 72 hr of incubation in the presence of DMSO, 2000 U · $mL^{-1}$ interferon-γ (ImmunoTools, Friesoythe, Germany) was added to the cell culture (*Tlili et al., 2011*). In a previous study, differentiation was checked by detecting the expression of the associated surface markers CD11b and CD64 (see *Degrossoli et al., 2018*). The viability of the cells was evaluated using trypan blue dye and was typically >90%.

## Chemiluminescence-based NADPH-oxidase activity assay

NADPH-oxidase-dependent superoxide production was selectively measured by chemiluminescence (CL) using the chemiluminogenic substrate lucigenin (10,10'-Dimethyl-9,9'-biacridinium dinitrate; Carl Roth, Karlsruhe, Germany) (*Minkenberg and Ferber, 1984*).

Differentiated PLB-985 cells were washed once with 1xPBS pH 7.4 and diluted in the same buffer to a final concentration of $5 \times 10^6$ cells · $mL^{-1}$. 100 µL of this cell suspension were placed in the wells of a non-transparent, white, clear-bottom 96-well plate (Nunc, Rochester, NY). For some experiments, cells were preincubated with 300 U/mL superoxide dismutase (SOD; Sigma-Aldrich, St. Louis) and/or 2000 U/mL catalase (Sigma-Aldrich), 10 µM diphenyleneiodium (DPI; NADPH oxidase inhibitor), 100 nM wortmannin (phosphoinositide 3-kinase (PI3K) inhibitor), 200 nM Gö 6983 (protein kinase C (PKC) inhibitor) or vehicle (1% DMSO) for 30 min at 37°C. All inhibitors dissolved in DMSO were purchased from Sigma-Aldrich. The final concentration of DMSO in all wells was adjusted to 1%. The bottom of the plate was covered using a white, non-transparent adhesive seal prior to measurement. 50 µL of either 1xPBS (resting CL) or agents to be tested, including the untreated and treated plasma proteins were added to the respective wells. Final concentrations were: 2 mg · $mL^{-1}$ α$_2$-macroglobulin, 3 mg · $mL^{-1}$ HSA, IgG, γ-globulins and Cohn fraction IV, 0.2 µM PMA (Phorbol 12-myristate 13-acetate; Sigma-Aldrich), 180.45 µM or 17.865 mM (i.e. the same or 99-fold higher concentration than HSA, respectively) taurine chloramine, based on the taurine concentration, as described above. Lucigenin was dissolved in 1xPBS to a concentration of 400 µM immediately prior to measurement and 50 µL were added to the wells. Chemiluminescence was measured every 1–2 min over 1.5 hr at 37°C using the Synergy H1 multi-detection microplate reader (Biotek, Bad Friedrichshall, Germany) in triplicates and chemiluminescence activity was expressed as integrated total counts as calculated by the addition of rectangles with unit width under individual data points. In inhibition assays, NADPH-oxidase activation, as measured by chemiluminescence activity, induced

by PMA, HSA, treated with a 50-fold molar excess of HOCl, or IgG, treated with a 150-fold molar excess of HOCl, was set to 100% and other relevant data as percentage of this control.

## Detection of reactive oxygen species using the fluorescence probe H₂DCFDA

Intracellular production of reactive oxygen species can be monitored by the oxidation of non-fluorescent 2',7'-dichlorodihydrofluorescein diacetate (H$_2$DCF-DA) to the fluorescent 2', 7'-dichlorofluorescein (DCF). H$_2$DCF-DA (Thermo Fisher Scientific, Waltham, MA) at a final concentration of 7 µM was preincubated with 1xPBS buffer in a non-transparent, black, clear-bottom 96-well plate (Nunc, Rochester, NY) for 15 min at 37°C. Fluorescence intensity was recorded every 2 min using the Synergy H1 multi-detection microplate reader (Biotek) at an excitation wavelength of 488 nm and an emission wavelength of 525 nm. Then 34 µM native or HOCl-treated serum albumin, or buffer were added to the wells. Fluorescence intensity was then recorded for 45 min.

## Cloning of Ag85B

*E. coli* strains, plasmids and primers used in this study are listed in **Table 1**. Ag85B gene (*fbpB*) of *Mycobacterium bovis* (differs by one base-pair from corresponding gene of *M. tuberculosis*, resulting in a Leu to Phe replacement at position 100; **De Wit et al., 1994**) was designed with optimized codon usage for expression in *E. coli*, synthesized and cloned into pEX-A2 standard vector by Eurofins Genomics. *fbpB* was amplified from pEXA_fbpB by PCR using primers fbpB-fw and fbpB-rv, purified according to the instructions of the NucleoSpin Gel and PCR Clean-up Kit (Macherey-Nagel GmbH, Düren, Germany) and cloned into the pET22b(+) expression vector via the restriction sites NdeI and XhoI with a hexahistidine (His$_6$)-tag placed at the C-terminal end of the gene. *E. coli* DH5α cells were transformed with the plasmid using a standard heat-shock method and plated on Luria Bertani (LB) agar plates supplemented with 50 µg · mL$^{-1}$ ampicillin. Screening for recombinant plasmids was performed by colony PCR, followed by isolation of potentially correct plasmids from the respective strains. Successful cloning of *fbpB* gene into pET22b(+) vector was verified by sequencing.

## Expression and purification of Ag85B

For heterologous expression and subsequent purification of His$_6$-tagged Ag85B, recombinant plasmid pET22b-fbpB was transferred into the *E. coli* expression strain BL21 (DE3) (see **Table 1**). The transformed cells were plated on LB agar plates containing 50 µg · mL$^{-1}$ ampicillin and incubated at 37°C for 24 hr. 2 × 50 ml LB medium with ampicillin were inoculated with a single colony from the agar plate and incubated at 37°C overnight. These overnight cultures were then used for inoculation of 5 × 1 L ampicillin-containing LB medium to a starting optical density at 600 nm (OD$_{600}$) of 0.1. The bacteria were grown at 37°C with shaking at 130 rpm until the OD$_{600}$ was ~0.5. Expression of

**Table 1.** *E. coli* strains, plasmids and primers used in this study.

| | Relevant properties or genotype | Source or reference |
|---|---|---|
| *E. coli* strains | | |
| DH5α | *supE44, ΔlacU169 (φ80 lacZΔM15) hsdR17 recA1 endA1 hsdR gyrA relA thi* | Invitrogen |
| BL21(DE3) | *F– ompT gal dcm lon hsdSB(rB- mB-) λ(DE3 [lacI lacUV5- T7 gene one ind1 sam7 nin5])* | Stratagene, Santa Clara, CA |
| Plasmids | | |
| pEXA_fbpB | Amp$^R$ pEX-A128 vector carrying synthesized *fbpB* gene from *M. bovis* | Eurofins Genomics |
| pET22b (+) | Amp$^R$, vector for overexpression of genes in *E. coli* | Novagen |
| pET22b-fbpB | Amp$^R$, vector for overexpression of *fbpB* gene in *E. coli* BL21(DE3) | This study |
| Primers | Sequence (5' - > 3') | |
| fbpB-fw | CCCCATATGTTCTCTCGTCCGG | |
| fbpB-rv | CCCCTCGAGACCAGCACCCAG | |

* Amp$^R$, ampicillin resistance.
DOI: https://doi.org/10.7554/eLife.47395.055

Ag85B including a C-terminal hexahistidine (His$_6$)-tag was induced with 1 mM isopropyl-beta-D-thio-galactopyranoside (IPTG) and continued for ~12 hr (overnight) at 20°C. Cells were harvested by centrifugation at 6500 x g for 45 min at 4°C. Pellets were washed once with 1xPBS and resuspended in lysis buffer (5 mM imidazole, 300 mM NaCl, 50 mM NaH$_2$PO$_4$, pH 8.0). Cells were disrupted using a Constant Cell disrupter (Constant Systems Limited, Daventry, England), and the obtained lysate was centrifuged for 45 min at 4°C and 57,500 x g. Solid precipitate and the supernatant were separated and evaluated for detection of recombinant protein by SDS-PAGE, followed by Coomassie Blue G-250 staining. Overexpressed Ag85B protein was found predominantly in the pellet fraction. Pellet was thus resuspended in lysis buffer containing 8 M urea and thoroughly homogenized. Suspension was applied to a polystyrene column filled with nickel-nitrilotriacetic (Ni-NTA) resin, and washed with urea-containing lysis buffer, followed by a washing step with wash buffer supplemented with a small amount of imidazole (8M urea, 20 mM imidazole, 300 mM NaCl, 50 mM NaH$_2$PO$_4$, pH 8.0). Recombinant Ag85B was eluted from the column using 250 mM imidazole buffer solution (8M urea, 250 mM imidazole, 300 mM NaCl, 50 mM NaH$_2$PO$_4$, pH 8.0). In order to remove imidazole, combined elution fraction was diluted 1:10 in sodium phosphate buffer (300 mM NaCl, 50 mM NaH$_2$PO$_4$, pH 8.0) and mixed overnight at 4°C. Precipitated protein was collected by centrifugation for 45 min at 4°C and 57,500 x g and resuspended in urea-containing sodium phosphate buffer (8M urea, 300 mM NaCl, 50 mM NaH$_2$PO$_4$, pH 8.0). Concentration of purified Ag85B protein was quantified spectrophotometrically by measuring the absorbance at 280 nm using a JASCO V-650 UV/VIS spectrophotometer. The molar extinction coefficient used for Ag85B was $\varepsilon_{280}$ = 75,860 M$^{-1}$ cm$^{-1}$ [108].

## Flow-cytometry-based cell apoptosis assay

Differentiated PLB-985 cells were counted, washed once with 1xPBS pH 7.4 and diluted in RPMI-1640 medium supplemented with 10% heat-inactivated FBS to a final concentration of $3.3 \times 10^6$ cells · mL$^{-1}$. 600 μL of this cell suspension were placed in the wells of a transparent, flat-bottom 24-well plate (Sarstedt, Nümbrecht, Germany). The cells were preincubated with 100 nM wortmannin, 50 μM Z-VAD-FMK, a broad-spectrum pan-caspase inhibitor (dissolved in DMSO; Santa Cruz Biotechnology, Dallas, TX), 250 μM cytochalasin D, an inhibitor of actin polymerization (dissolved in DMSO; Sigma-Aldrich, St. Louis), 155 μM native or modified HSA, that has been treated with a 50-fold molar excess of HOCl as described above, or vehicle (1% DMSO) for 1 hr at 37°C. The final concentration of DMSO in all wells was adjusted to 1%. Subsequently, 1 μM Ag85B (85 μM stock in urea-containing buffer (8 M urea, 300 mM NaCl, 50 mM NaH$_2$PO$_4$, pH 8.0)), 2 μM staurosporine (dissolved in DMSO; Sigma-Aldrich) or vehicle (94 mM urea and 1% DMSO) were added. After 1 hr of incubation with Ag85B or 6 hr with staurosporine at 37°C, cells were washed once with cold 1xPBS, followed by one washing step with 1x Annexin V binding buffer. For the analysis of cell viability, the variously treated cell suspensions were stained with Annexin V-FITC and propidium iodide (PI) using the Dead Cell Apoptosis Kit (Invitrogen, Thermo Fisher Scientific, Waltham, MA) according to the manufacturer's instructions and subsequently subjected to flow cytometry. Samples were analyzed using a BD FACSCanto II flow cytometer (Becton, Dickinson and Company). Fluorescence emitted by Annexin V-FITC and PI was measured through a 530/30 nm and 585/42 nm bandpass filter, respectively, upon excitation with an argon ion laser operating at 488 nm. Single-stained compensation controls were used to calculate the compensation matrix. 20,000 events were acquired and recorded per sample. Data were analyzed using FlowJo (version 10) software.

## Fluorescent labeling of Ag85B

For fluorescent labeling of Ag85B the green fluorescent dye CF 488A succinimidyl ester (Sigma-Aldrich, St. Louis) was used. Labeling occurred via reaction of the succinimidyl ester group of the dye with amine groups of Ag85B. Labeling was performed according to the manufacturer's instructions with slight modifications. To prevent aggregation of Ag85B, labeling reaction was performed in urea-containing buffer (8 M urea, 300 mM NaCl, 50 mM NaH$_2$PO$_4$, pH 8.0) and carried out for 6 hr at room temperature under continuous shaking. Excess dye was removed by size-exclusion chromatography using PD-10 desalting columns containing Sephadex G-25 resin according to the manufacturer's instructions (GE Healthcare Life Sciences, Amersham, UK). Concentration of the conjugate and degree of labeling (DOL) were calculated using the formula provided by the dye's manufacturer. DOL was 3.1.

## Ag85B-488 uptake assay

For the antigen uptake assay, differentiated PLB-985 cells were incubated with fluorescently-labeled Ag85B (Ag85B-488) protein in the presence or absence of native and HOCl-modified HSA in the same way as for the cell apoptosis assay described above. After 1 hr of incubation at 37°C, cells were washed twice with 1xPBS, followed by fixation with 4% paraformaldehyde (PFA) for 10 min on ice. Cells were washed with 1xPBS and subsequently subjected to flow cytometry. Samples were analyzed using a BD FACSCanto II flow cytometer. Fluorescence emitted by Ag85B-488 was measured through a 530/30 nm bandpass filter upon excitation with the 488 nm argon laser line. 30,000 events were acquired and recorded per sample. Data were analyzed using FlowJo (version 10) software.

## Ag85B aggregation assay

Aggregation of Ag85B was induced by the stepwise addition of 75 µL Ag85B stock (5 × 15 µL every 20 s) to 1525 µl 40 mM HEPES/KOH buffer, pH 7.5. Final concentration of Ag85B in the aggregation assay was 0.186 µM. Native or variously treated HSA as described above, was added to the assay buffer to a final concentration of 14.88 µM (corresponding to a 80-fold molar excess over Ag85B) prior to the addition of Ag85B. The increase of light scattering was monitored in a JASCO FP-8500 fluorescence spectrometer equipped with an EHC-813 temperature-controlled sample holder at 30°C for 200 s under continuous stirring. Measurement parameters were set to 360 nm (Ex/Em), 2.5 nm slit width (Ex/Em) and medium sensitivity. Chaperone activity was expressed as the difference between initial and final light scattering of an individual sample in arbitrary units. Aggregation of Ag85B in the absence of HSA was set to 100%.

## Acknowledgements

Funding for this study was provided by the German Research Foundation (DFG) through grant LE2905/1-2 to LIL as part of the priority program 1710 'Dynamics of Thiol-based Redox Switches in Cellular Physiology'. Parts of this manuscript were written during a Writing Retreat funded by the Ruhr-Universität Bochum Research School RURS[plus].

## Additional information

### Funding

| Funder | Grant reference number | Author |
|---|---|---|
| Deutsche Forschungsgemeinschaft | LE2905/1-2 | Lars I Leichert |

The funders had no role in study design, data collection and interpretation, or the decision to submit the work for publication.

### Author contributions

Agnes Ulfig, Conceptualization, Formal analysis, Investigation, Visualization, Methodology, Writing—original draft, Writing—review and editing; Anton V Schulz, Conceptualization, Formal analysis, Investigation, Methodology, Writing—review and editing; Alexandra Müller, Conceptualization, Investigation, Methodology; Natalie Lupilov, Investigation, Methodology; Lars I Leichert, Conceptualization, Supervision, Funding acquisition, Writing—review and editing

### Author ORCIDs

Lars I Leichert (ID) https://orcid.org/0000-0002-5666-9681

### Decision letter and Author response

Decision letter https://doi.org/10.7554/eLife.47395.058
Author response https://doi.org/10.7554/eLife.47395.059

## Additional files

### Supplementary files
• Transparent reporting form
DOI: https://doi.org/10.7554/eLife.47395.056

### Data availability
Source data generated or analysed during this study are included in the manuscript and supporting files. Source data files have been provided for Figures 1 to 9 and supplementary figures where appropriate.

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
