## [Decision Letter]

Thank you for submitting your article "*N*-chlorination mediates protective and immunomodulatory effects of oxidized human plasma proteins" for consideration by *eLife*. Your article has been reviewed by three peer reviewers, and the evaluation has been overseen by a Reviewing Editor and Satyajit Rath as the Senior Editor. The following individual involved in review of your submission has agreed to reveal their identity: Tony Kettle (Reviewer #1).

The reviewers have discussed the reviews with one another and the Reviewing Editor has drafted this decision to help you prepare a revised submission.

Summary:

This manuscript reports on the role of HOCl-modification of HSA and other plasma proteins and the potential role of the resulting N-chlorinated materials as chaperones that can prevent the aggregation of other proteins, act as survival signaling molecules for activated neutrophils and also as stimulators of neutrophil oxidant production. The paper is very interesting, with potentially important findings.

Essential revisions:

1) The major concern is the concentration of HOCl used to treat the proteins investigated. The authors use very high levels of HOCl – up to 50 mM, and justify these levels referring to Summers et al., 2008. However, we did not find definitive information in this paper for concentrations of > 15 mM in plasma and at sites of inflammation. Furthermore the levels quoted, are for the concentration of HOCl generated over a long period, rather than a bolus addition as used here. Thus, 50 mM seems extreme and may be misleading. It is questionable that at sites of inflammation, oxygen supply could be maintained to fuel HOCl production. Rather, it is likely that at such rates of HOCl production, sites of inflammation would become rapidly anaerobic.

At the doses used in the current study, there will be high levels of chlorination of tyrosine residues in human serum albumin (Kettle FEBS Letters 1996). These levels are well above what is found in proteins in lavage fluid from children with cystic fibrosis, which have the highest reported levels of 3-chlorotyrosine. Thus the authors should justify the doses of HOCl used and use a more extensive dose range with respect to test the proposed functions of chlorinated proteins.

2. Also relating to the dose of HOCl used, is that the levels of chloramines on the proteins are likely to be very high. However, these have not been measured in the current work and that is a clear weakness. The authors should measure the concentration of protein chloramines formed when proteins are treated with HOCl (see Dypbukt et al., 2005).

3) Given the likely high concentration of protein chloramines in this study, it is possible that chloramine reactivity is responsible for the observed effects. Therefore, the authors should use taurine chloramine as a control to establish whether the effects they observe are due to protein chloramines or simply just chloramines.

4) With respect to the chaperone activity of chlorinated proteins, it would be prudent to use another method for following protein aggregation since it is possible that chloramine reactivity affects light scattering in ways other than preventing aggregation of proteins.

5) With respect to the stimulation of oxidant production by neutrophil-like cells, it is possible that the observed effects are due to the reactivity of added chloramines with the probes used to measure oxidant production. Therefore, in every instance the authors should use taurine chloramine as a control to demonstrate that minimal signal is coming from the reaction of chloramine with their probe.

6) To demonstrate that protein chloramines activate the NADPH oxidase in cells, in every case, controls should be performed whereby cells are treated with DPI, superoxide dismutase, catalase, or superoxide dismutase and catalase. These controls should establish whether oxidation of the probe results from activation of the NADPH oxidase and subsequent production of superoxide and hydrogen peroxide.

7) Another major concern is that the authors fail to unambiguously demonstrate that HOCl turns serum proteins into chaperones via the N-chlorination of positively charged residues. This conclusion is based, in part, on the fact that serum proteins lose their chaperone activity following treatment with ascorbate, an antioxidant that "typically does not reduce native or HOC-induced disulfide bond". However, this is not shown directly. And no reference is provided to support that "the antioxidant ascorbate specifically removes N-chlorination". The authors also base their conclusions on the fact that methylation of basic residues prevents the HOCl-induced activation. However, given the importance that N-chlorination plays in this paper, direct evidence must be provided, for instance using mass-spectrometry to identify the N-chlorinated residues and to show the reversibility of this modification by ascorbate.

What is the evidence that it does not reduce HOCl-induced disulfide bonds?

8) The authors assume that ascorbate and Met remove all the chloramines formed, but do not measure this. Chloramines are relatively easy to quantify, so it is not clear why this has not been done (see for example the work of Hawkins with proteins, and also plasma). A decomposed chloramine control should also be examined (in addition to the reduced control) as the decomposed species (carbonyls etc.) may also have effects.

9) It would also be interesting to include more control experiments, for instance to show that exposing albumin or other plasma proteins to oxidants such as H_2_O_2_ does not mimic treatment with HOCl, and to show that the N-chlorinated proteins work as chaperones for at least a second substrate (and not only with citrate synthase).

10)The authors assume that ascorbate only removes N-chlorinated species on the HOCl-modified proteins. Other transients such as sulfenyl chlorides and sulfenic acids, which are also formed by HOCl, are also reactive with ascorbate.

11) There are likely to be significant modifications of other residues such as disulfides, Trp and Tyr residues on the modified proteins. Some of the species may also give rise to reducible species (e.g. N-Cl species formed at the indolyl nitrogen of Trp).

12) Regarding the experimental part on neutrophils: why were the experiments not carried out on fresh neutrophils from blood? Alternatively, but less ideal would be testing different myeloid cell lines to show that the effect is not specific of the PLB-985 cell line.

13) The method used to generate the methylated proteins is not referenced or validated, and there is no assessment of the extent of modification of the basic residues present on the proteins. This is essential information, as is evidence that only Lys and Arg are affected.

14) A final major issue is that the statistical analyses are inappropriate: Student's t-tests as used here are not suited for multiple comparisons.

---

## [Author Response]

Essential revisions:1) The major concern is the concentration of HOCl used to treat the proteins investigated. The authors use very high levels of HOCl – up to 50 mM, and justify these levels referring to Summers et al., 2008. However, we did not find definitive information in this paper for concentrations of > 15 mM in plasma and at sites of inflammation. Furthermore the levels quoted, are for the concentration of HOCl generated over a long period, rather than a bolus addition as used here. Thus, 50 mM seems extreme and may be misleading. It is questionable that at sites of inflammation, oxygen supply could be maintained to fuel HOCl production. Rather, it is likely that at such rates of HOCl production, sites of inflammation would become rapidly anaerobic.At the doses used in the current study, there will be high levels of chlorination of tyrosine residues in human serum albumin (Kettle FEBS Letters 1996). These levels are well above what is found in proteins in lavage fluid from children with cystic fibrosis, which have the highest reported levels of 3-chlorotyrosine. Thus the authors should justify the doses of HOCl used and use a more extensive dose range with respect to test the proposed functions of chlorinated proteins.

We thank the reviewers for their comment. We now tested a more extensive dose range with concentrations suggested by the reviewers. At concentrations of 0.3 mM HSA (as they are expected in the interstitial space) and 15 mM HOCl concentration, we still saw full activation. The same is true for even lower concentrations of HSA and HOCl, as long as the molar excess of HOCl is between 10- to 50-fold. As HSA, by sheer abundance, is probably a major sink for HOCl, we would like to propose that these experiments mimic an inflammatory situation where the presence of a large number of neutrophils would lead to high local HOCl concentrations in a confined space. Without significant circulation in this space and thus no appreciable exchange of serum proteins, over time the serum proteins such as HSA could be exposed to these HOCl concentrations. This new data is now presented in Figure 2A and B and subsection “Albumin, the major protein component of serum, shows HOCl-induced chaperone-like activity” We hope this satisfies the reviewers.

2) Also relating to the dose of HOCl used, is that the levels of chloramines on the proteins are likely to be very high. However, these have not been measured in the current work and that is a clear weakness. The authors should measure the concentration of protein chloramines formed when proteins are treated with HOCl (see Dypbukt et al., 2005).

Although we indirectly determined N-chlorination by determining the reversible loss of amino groups using a fluorescamine assay (see Figure 3A and B), we have now determined the chloramines directly using the assay described by Dypbukt et al. These results have been added to Figure 3C and D and the first paragraph of the subsection “Decreased amino group content of HSA upon HOCl treatment is accompanied by an increased overall hydrophobicity of the protein”.

3) Given the likely high concentration of protein chloramines in this study, it is possible that chloramine reactivity is responsible for the observed effects. Therefore, the authors should use taurine chloramine as a control to establish whether the effects they observe are due to protein chloramines or simply just chloramines.

We have now performed the requested control experiments. Neither TauNCl nor TauNCl_2_ at concentrations corresponding up to the maximum number of potentially reversible N-chlorination sites in HSA had any influence on the chaperone or immune cell activation assay. These results are now briefly mentioned in the first paragraph of the subsection “Decreased amino group content of HSA upon HOCl treatment is accompanied by an increased overall hydrophobicity of the protein” and in the fifth paragraph of the subsection “Activation of neutrophil-like cells by HOCl-treated serum albumin is based on reversible N-chlorination”, and have been added to Figure 2—figure supplement 4 and Figure 4—figure supplement 3, but we are perfectly happy to add them to the main manuscript, if the reviewers or editors deem it appropriate.

4) With respect to the chaperone activity of chlorinated proteins, it would be prudent to use another method for following protein aggregation since it is possible that chloramine reactivity affects light scattering in ways other than preventing aggregation of proteins.

With the control experiments using TauNCl mentioned above, we now show that chloramine reactivity does not affect our light scattering experiments. We hope that satisfies the reviewers.

5) With respect to the stimulation of oxidant production by neutrophil-like cells, it is possible that the observed effects are due to the reactivity of added chloramines with the probes used to measure oxidant production. Therefore, in every instance the authors should use taurine chloramine as a control to demonstrate that minimal signal is coming from the reaction of chloramine with their probe.

We had already tested this possibility by adding HSA_HOCl_ as a chloramine directly to Lucigenin in the absence of cells (Figure 4B). We mentioned this in the Results section and showed that H_2_DCF-DA, another probe often used in such experiments, indeed reacts with HSA_HOCl_ in the supplements (now Figure 4 —figure supplement 2). To further clarify, we now also refer to the above mentioned TauNCl control experiment in the fifth paragraph of the subsection “Activation of neutrophil-like cells by HOCl-treated serum albumin is based on reversible N-chlorination”,.

6) To demonstrate that protein chloramines activate the NADPH oxidase in cells, in every case, controls should be performed whereby cells are treated with DPI, superoxide dismutase, catalase, or superoxide dismutase and catalase. These controls should establish whether oxidation of the probe results from activation of the NADPH oxidase and subsequent production of superoxide and hydrogen peroxide.

We already did the suggested experiment in the presence of DPI (see Figure 7), but we now also do the control experiments in the presence of SOD, catalase, and both SOD and catalase. The experiments show the expected results and are now presented in Figure 7—figure supplement 1 and mentioned in the fourth paragraph of the subsection “Activation of NADPH oxidase by N-chlorinated serum albumin and immunoglobulin G occurs predominantly via PI3K-dependent signaling pathways”. As above, we are perfectly happy to add the figures to the main manuscript, if the reviewers or editors deem it appropriate.

7) Another major concern is that the authors fail to unambiguously demonstrate that HOCl turns serum proteins into chaperones via the N-chlorination of positively charged residues. This conclusion is based, in part, on the fact that serum proteins lose their chaperone activity following treatment with ascorbate, an antioxidant that "typically does not reduce native or HOC-induced disulfide bond". However, this is not shown directly. And no reference is provided to support that "the antioxidant ascorbate specifically removes N-chlorination". The authors also base their conclusions on the fact that methylation of basic residues prevents the HOCl-induced activation. However, given the importance that N-chlorination plays in this paper, direct evidence must be provided, for instance using mass-spectrometry to identify the N-chlorinated residues and to show the reversibility of this modification by ascorbate. What is the evidence that it does not reduce HOCl-induced disulfide bonds?

Full-length MS with such large proteins is not feasible in our lab. HSA is particularly challenging, as it contains a number of different modifications and thus does not easily resolve into a “single” peak, as e.g. the much smaller RidA does. MS of peptides is complicated by the instability of the modification in question (standard proteolytic digestion protocols typically include DTT as a reducing agent) and the fact that Arg and Lys are modified, both amino acids recognized and required for trypsin cleavage, the most commonly used protease in MS. We are currently working on an MS labeling approach, but we feel that this goes beyond the scope of this manuscript. The effects that these modifications have, are reversible by ascorbate (Figures 1, 2, 3, 4, 5, 6), can be prevented by prior methylation of amino groups (Figures 1, 2, 3, 4, 5, 6), elevate the hydrophobicity of the protein (Figure 3), we now show that they are additionally reversible by methionine (Figure 3, Figure 2—figure supplement 2), we now show that HOCl-treated HSA indeed contains N-chloramines (Figure 3C, D), we additionally demonstrate that no appreciable amount of intermolecular disulfides is formed between the sole accessible thiol and that DTT reduces intermolecular disulfides, while ascorbate does not under the conditions we used in our experiments (Figure 2—figure supplement 3). Unless the reviewers give us a specific hypothesis that we could test against, we cannot come up with any modification other than N-chlorination of positively charged residues that could explain the observed effects. Our new results are mentioned throughout the manuscript.

8) The authors assume that ascorbate and Met remove all the chloramines formed, but do not measure this. Chloramines are relatively easy to quantify, so it is not clear why this has not been done (see for example the work of Hawkins with proteins, and also plasma). A decomposed chloramine control should also be examined (in addition to the reduced control) as the decomposed species (carbonyls etc.) may also have effects.

We now performed the suggested experiment and determined the amount of chloramines in HSA after treatment with HOCl and after reduction with ascorbate (see above). We hope this convinces the reviewers that indeed almost all chloramines are removed by ascorbate. We feel that testing effects of decomposition species is beyond the scope of our study. However, we think that our data shows, that decomposition species are either not formed at significant amounts in HSA under the conditions tested (as suggested by the data in Figure 3C and D: full restoration of amino groups by ascorbate reduction, no detectable chloramines left), or, if they are formed and we misinterpret the data, they do not play a role in the physiological effects observed: both chaperone activity and immune cell activation are reversed by ascorbate (Figures 1, 4). Decomposition products should not be reverted by ascorbate, and if they were present and/or had an effect, we should see activity, but we do not.

9) It would also be interesting to include more control experiments, for instance to show that exposing albumin or other plasma proteins to oxidants such as H_2_O_2_ does not mimic treatment with HOCl, and to show that the N-chlorinated proteins work as chaperones for at least a second substrate (and not only with citrate synthase).

We have now tested hydrogen peroxide, it does not mimic HOCl treatment (Figure 2E and F). We had already tested chaperone activity with Ag85B (Figure 9A and B), but we now also include IlvA, another substrate that works in chaperone assays (Müller et al., 2014) (Figure 2—figure supplement 1). We mention these new results in the subsection” Albumin, the major protein component of serum, shows HOCl-induced chaperone-like activity”.

10)The authors assume that ascorbate only removes N-chlorinated species on the HOCl-modified proteins. Other transients such as sulfenyl chlorides and sulfenic acids, which are also formed by HOCl, are also reactive with ascorbate.

While sulfenyl chlorides and sulfenic acids could react with ascorbate, we do not think that this is the case in our experiments for several reasons: both of those modifications are highly unstable and it is not likely that they would exist for as long as our experiments last. Additionally, in HSA, with which we performed most of the experiments, there is only one cysteine available that could potentially form such a modification (all other cysteines are engaged in structural disulfides), however, to see meaningful effects, we need a certain molar excess of HOCl. Furthermore, N-methylation would not prevent these modifications from happening and we should still see activation of the proteins by HOCl. We however now discuss this possibility in the subsection “Methylation of basic amino acid residues in serum albumin inhibits HOCl-induced activation of its chaperone function”.

11) There are likely to be significant modifications of other residues such as disulfides, Trp and Tyr residues on the modified proteins. Some of the species may also give rise to reducible species (e.g. N-Cl species formed at the indolyl nitrogen of Trp).

As the reviewers point out, most of these modifications are irreversible. We do not argue that they do not exist, we just argue that they are mostly irrelevant: reduction by ascorbate removes the effects pointing clearly to a reversible modification. We do not observe appreciable amounts of other hypothetically reversible modifications such as disulfide bonds on a non-reducing gel (Figure 2—figure supplement 3). We now discuss that Nitrogen-Heteroatoms in the organic ring systems of Trp and His could potentially be chlorinated as well (subsection “Decreased amino group content of HSA upon HOCl treatment is accompanied by an increased overall hydrophobicity of the protein”). However, in case of tryptophan, reactions with HOCl seem to favor the C3 or C7 position in the ring and not the nitrogen (Fu et al. 2006, “Specific Sequence Motifs Direct the Oxygenation and Chlorination of Tryptophan by Myeloperoxidase” Biochemistry 45(12): 3961-3971, Dong et al. 2005 “The structure of tryptophan 7-halogenase (PrnA) suggests a mechanism for regioselective chlorination” Science 309 (5744): 2216-2219). Both His and Trp are present in much lower abundance in HSA.

12) Regarding the experimental part on neutrophils: why were the experiments not carried out on fresh neutrophils from blood? Alternatively, but less ideal would be testing different myeloid cell lines to show that the effect is not specific of the PLB-985 cell line.

We currently do not have an ethics vote on experiments with human specimens. However, these experiments have essentially been done in other studies (although without providing a mechanism for the observations), which we mention in our Introduction.

13) The method used to generate the methylated proteins is not referenced or validated, and there is no assessment of the extent of modification of the basic residues present on the proteins. This is essential information, as is evidence that only Lys and Arg are affected.

We thank the reviewers for pointing out our oversight. We now provide the reference (Rypniewski et al., 1993) in the subsection “Methylation of proteins”. Protein N-Methylation is a method widely used in crystallography in order to be able to crystallize challenging proteins. There are commercial kits based on this methodology. As such, selective methylation was often verified in crystallographic studies based on the X-ray structure. We hope this convinces the reviewers.

14) A final major issue is that the statistical analyses are inappropriate: Student's t-tests as used here are not suited for multiple comparisons.

We want to clarify that we only apply the t-test to compare two groups in each instance (the two groups bracketed in the bar graphs). This was not done post-hoc but very much hypothesis driven (i.e. we did not randomly perform experiments and then performed t-tests over all groups and then cherry-picked “significant” results). As an example, Figure 4C: as we assumed that N-chlorination is the modification causing the effects, we expected that ascorbate reduction would reverse the effects, so we performed a t-test comparing groups “HSA _50xHOCl_” and “HSA _50xHOCl/Asc_”. As prior methylation of basic amino acids should have the same effect as ascorbate reduction, we did not compare the two groups “HSA _50xHOCl/Asc_” and “HSA _Met/50xHOCl_”. Thus, we feel our use of the t-test is appropriate and others concur: Rothmann 1990, “No adjustments are needed for multiple comparisons.” Epidemiology 1(1): 43-46. We hope this satisfies the reviewers. Using a large part of the scientific literature as our guide, we found that our use of the t-test is very much customary, thus we currently do not justify it in the manuscript, however, we are happy to add a justification (e.g. above mentioned reference) to the manuscript, if the reviewers or editors deem it necessary.